# Understanding the Insulin-Degrading Enzyme: A New Look at Alzheimer’s Disease and Aβ Plaque Management

**DOI:** 10.3390/ijms26146693

**Published:** 2025-07-12

**Authors:** Michele Cerasuolo, Maria Chiara Auriemma, Irene Di Meo, Carmen Lenti, Michele Papa, Giuseppe Paolisso, Maria Rosaria Rizzo

**Affiliations:** 1Department of Advanced Clinical and Surgical Sciences, University of Campania “Luigi Vanvitelli”, 80138 Naples, Italy; michele.cerasuolo@unicampania.it (M.C.); auri.mariachiara@gmail.com (M.C.A.); irene.dimeo@unicampania.it (I.D.M.); carmen.lenti@unicampania.it (C.L.); giuseppe.paolisso@unicampania.it (G.P.); 2Laboratory of Neuronal Networks Morphology and System Biology, Department Mental and Physical Health and Preventive Medicine, University of Campania “Luigi Vanvitelli”, 80138 Naples, Italy; michele.papa@unicampania.it; 3Saint Camillus International University of Health and Medical Sciences (UniCamillus), 00131 Rome, Italy

**Keywords:** insulin-degrading enzyme (IDE), type 2 diabetes mellitus (T2DM), insulin resistance, Alzheimer’s disease (AD), amyloid-β (Aβ) plaques

## Abstract

Insulin-degrading enzyme (IDE) plays a critical role in regulating insulin levels in various tissues, including the brain, liver, and kidneys. In type 2 diabetes mellitus (T2DM), key features include insulin resistance, elevated insulin levels in the blood, and hyperglycemia. In this context, the function of IDE becomes particularly important; however, in T2DM, IDE’s function can be impaired. Notably, individuals with T2DM have a higher risk of developing Alzheimer’s disease (AD), suggesting that impaired IDE function may contribute to both diabetes and neurodegeneration. IDE has been studied for its ability to degrade Amyloid-β peptides, the primary constituents of amyloid plaques in AD. However, its role in Aβ clearance in vivo remains debated due to limited enzymatic efficacy under physiological conditions and differences in subcellular localization between IDE and its putative substrate. Other proteases, such as neprilysin, appear to play a more prominent role in preventing plaque formation. Additionally, the long-standing hypothesis that insulin competes with Aβ for IDE activity has been questioned, as brain insulin levels are too low to inhibit Aβ degradation significantly. Genetic variants in the IDE gene have been associated with increased AD risk, although the mechanisms by which they alter enzyme function are not yet fully understood. A deeper understanding of IDE’s role in the context of both metabolic and neurodegenerative diseases may provide valuable insights for the development of new therapeutic strategies.

## 1. Introduction

In Type 2 diabetes (T2DM), insulin resistance in peripheral tissues, abnormalities in insulin signaling pathways, and hyperinsulinemia are key contributors to hyperglycemia [1]. In order to improve glucose uptake, numerous studies have focused on elucidating the underlying mechanisms of insulin resistance as well as exploring potential therapeutic interventions, both pharmacological and non-pharmacological [2]. Moreover, numerous studies have suggested a potential shared pathophysiology between T2DM and Alzheimer’s disease (AD). As a result, there has been growing interest in recent years in research aimed at deciphering the interplay mechanisms between the two conditions, including the possible role of antidiabetic drugs and their therapeutic potential in AD and related neurodegenerative disorders [3,4].

However, the insulin-degrading enzyme (IDE) plays a critical role in regulating insulin levels by ensuring that insulin does not remain active in the body longer than necessary [5]. Therefore, if IDE’s function is compromised, excess insulin can accumulate in the bloodstream, worsening insulin resistance, disrupting glucose metabolism, and contributing to the progression of diabetes [6].

Scientific research suggests that reduced IDE activity may lead to increased insulin resistance, while enhanced IDE activity could provide protective effects against it [7]. Additionally, IDE is essential for breaking down several proteins, including islet amyloid polypeptide (IAPP) [8], glucagon [9], insulin-like growth factors (IGF-1 and IGF-2) [10,11], and Amyloid-β (Aβ) [11].

However, it is important to note that many of these reported substrates have been primarily identified through in vitro assays or cell-free systems, which may not accurately reflect physiological relevance [12,13]. While IDE has been implicated in Aβ catabolism, the idea that increasing IDE levels in the central nervous system (CNS) would directly enhance Aβ clearance and improve cognitive function remains speculative. In vivo evidence does not fully support this hypothesis. Moreover, IDE exhibits high substrate promiscuity and often shows greater affinity for peptides such as insulin and amylin than for Aβ [14,15]. This raises concerns that augmenting IDE activity could interfere with other essential regulatory pathways, particularly those related to insulin signaling. IDE may contribute to the regulation of Aβ accumulation in AD and diabetic cognitive impairment. While some studies have proposed that increasing IDE levels in the central nervous system (CNS) might facilitate Aβ degradation and offer cognitive benefits, this hypothesis remains under investigation. Further exploration is required to determine the extent to which IDE influences these processes and whether modulating its activity could have therapeutic implications [11].

## 2. Ide Activity

IDE is a metalloprotease enzyme encoded by a gene on human chromosome 10 (q23-q25) and a gene on mouse chromosome 19 [16]. Structurally, it consists of four homologous domains (domains 1 and 2 form the N-terminal portion, while domains 3 and 4 form the C-terminal portion), connected by a short “hinge” loop. In addition to its primary function in insulin degradation, IDE is evolutionarily conserved, with homologs identified in organisms ranging from bacteria to eukaryotes, suggesting that similar enzymatic functions have been preserved over time [17,18].

However, recent evidence highlights that, despite its widespread conservation, IDE has undergone significant evolutionary adaptations. Specifically, bacterial IDE homologs often contain signal peptides that direct them toward the secretory pathway, whereas eukaryotic IDE typically lacks such sequences and localizes predominantly in the cytosol. This shift in subcellular localization is thought to reflect a functional diversification beyond insulin degradation. In microglia, for instance, IDE dynamically localizes to multivesicular bodies and is released in extracellular vesicles upon activation, suggesting broader roles in intercellular signaling and proteostasis regulation [19].

IDE is ubiquitously expressed, both in insulin-sensitive and non-insulin-sensitive cells, supporting a multifunctional role for this protein [20].

It is mainly localized in the cytoplasm of different cell types in various organs, including the brain, heart, liver, pancreas, and muscle [21]. Additionally, IDE can be secreted via exosomes, entering the extracellular space to interact with insulin and other substrates [22]. IDE is a zinc-dependent metalloprotease characterized by broad substrate specificity, and it plays key roles in the degradation of bioactive peptides such as insulin, amyloid-β, islet amyloid polypeptide, and glucagon. Its enzymatic activity is regulated by factors such as substrate conformation, subcellular compartmentalization, and interaction with small-molecule effectors. Recent studies have also identified several pharmacological modulators of IDE, including both inhibitors and activators; however, their substrate selectivity and potential clinical applications remain under active investigation [23,24,25]. Including these aspects highlights the complex structure–function relationship of IDE and its relevance beyond insulin metabolism.

### 2.1. IDE and Insulin Target

IDE has a strong affinity for insulin, and when its expression is increased, it can lead to lower insulin levels, potentially resulting in elevated blood glucose concentrations. Conversely, reducing IDE levels can limit the breakdown of insulin, which may help maintain more stable glycemia. However, the physiological role of IDE in insulin degradation remains a matter of ongoing debate [15]. While early studies supported a major role for IDE in hepatic insulin clearance, more recent findings have challenged this classical view. For instance, some studies have shown that mice lacking IDE display elevated plasma insulin levels, suggesting reduced insulin clearance. However, others report no significant difference in insulin levels between IDE-deficient and wild-type animals, indicating compensatory mechanisms may be in place [15].

Experimental support for IDE’s involvement in insulin degradation includes the identification of insulin fragments in cells that match those produced by IDE-mediated cleavage in vitro. Furthermore, the use of nonspecific IDE inhibitors, such as N-ethylmaleimide and bacitracin, has been shown to impair insulin internalization and degradation in multiple cell models [5,15,26,27]. Yet, as IDE is primarily localized in the cytosol and lacks a classical signal peptide, the exact cellular mechanisms that allow IDE to access insulin and other substrates remain incompletely understood [15]. According to recent analyses [28], this uncertainty is compounded by IDE’s unconventional subcellular distribution. In fact, IDE has been detected not only in the cytosol but also associated with membranes in hepatocytes [26], skeletal muscle cells [29], neurons, and microglia [19]. In these systems, IDE appears to partition between a cytosolic pool—with a relatively long half-life—and a membrane-associated pool characterized by a faster turnover.

Notably, although some studies reported IDE on the external cell surface [30,31,32], recent findings suggest that IDE localizes predominantly to the cytosolic face of membranes rather than being truly exposed at the cell surface. This is consistent with bioinformatic predictions indicating the absence of a signal peptide in IDE, which would preclude conventional secretion or membrane insertion [19]. Furthermore, a small fraction of IDE has been observed in specialized membrane microdomains such as lipid rafts, which may influence substrate accessibility and enzymatic function [30,31,32].

A significant issue regarding insulin degradation by IDE is identifying the cellular location where this occurs. A two-phase model has been proposed: one hypothesis suggests that membrane-associated IDE contributes to insulin degradation prior to internalization. However, in hepatocytes, evidence indicates that insulin bound to its receptor is internalized via endocytosis and delivered to endosomes, where it is degraded by various proteases, including cathepsin D, neutral aminopeptidase, and possibly IDE [19,33,34,35,36,37]. Once internalized, insulin is either recycled or directed toward degradation compartments, including lysosomes [38].

IDE may participate in the degradation of insulin within endosomes, contributing to the regulation of insulin availability and preventing prolonged insulin action that could promote insulin resistance and impair glucose metabolism [5]. However, this hypothesis is not without controversy. IDE is known to have optimal catalytic activity at neutral to slightly basic pH (approximately 7.3–8.5), whereas the pH of endosomal and lysosomal compartments—particularly late endosomes and lysosomes—ranges between 5.0 and 6.0. This acidic environment is suboptimal for IDE function, and several studies have demonstrated a marked reduction in its enzymatic efficiency at lower pH values [12]. These observations raise questions about the extent to which IDE contributes to insulin degradation in such acidic intracellular compartments. It has also been proposed that IDE may interact functionally with lysosomal proteases during endosomal–lysosomal fusion, potentially supporting insulin catabolism under specific conditions [5]. Moreover, IDE can degrade insulin’s A and B chains separately, generating fragments with biological activities that differ from the intact hormone, which may add an additional layer of complexity to its regulatory role in insulin metabolism [39].

### 2.2. IDE and Non-Insulin Target

IDE is capable of cleaving peptides that are up to 70 amino acids long, but it cannot process larger proteins [25]. In addition to its role in degradation, IDE performs several regulatory functions, including the modulation of androgen and glucocorticoid receptors, participating in peroxisomal fatty acid oxidation, facilitating antigen presentation, supporting cellular growth and differentiation, as well as being involved in proteasomal degradation [15].

While IDE primarily targets insulin as its preferred substrate, it can also cleave other short polypeptides, including glucagon, amylin, atrial natriuretic peptide, IGF-1 and IGF-2, and Aβ [40,41]. This enzymatic versatility has prompted interest in pharmacologically modulating IDE activity to influence metabolic homeostasis, particularly in the context of insulin-related disorders. In a high-fat diet (HFD)-induced preclinical model of type 2 diabetes, administration of the IDE inhibitor 6bK did not emerge as the main focus of the study. Instead, the key finding was that treatment with the IDE activator PIF enhanced IDE activity in pancreatic β-cells, resulting in increased insulin secretion during an intraperitoneal glucose tolerance test [42].

### 2.3. IDE and Amyloid-β Target

Aβ is a peptide that can accumulate abnormally in the brain in various conformations, including monomers, oligomers, fibrils, and extracellular plaques. These different Aβ species contribute to AD pathology by disrupting neuronal communication and triggering neuroinflammatory responses, ultimately leading to neurodegeneration [43].

IDE is one of the enzymes responsible for breaking down Aβ [44]. It is found in the brain and is believed to help clear Aβ from the extracellular space [45]. Elevated levels of IDE in the inactive, S-nitrosylated form have been found in the brains of individuals with AD [18]. While its involvement in insulin regulation is relatively well-characterized, its role in Aβ metabolism remains less clearly defined. The precise mechanisms by which IDE contributes to Aβ clearance, and the extent of its impact on amyloid plaque formation in the brain, are still under investigation [46,47]. Furthermore, because both insulin and Aβ are substrates for IDE, they may compete for binding and degradation. As a result, elevated insulin levels—such as those seen in hyperinsulinemic states—could reduce IDE’s availability for Aβ degradation, thereby impairing Aβ clearance from the brain [48,49]. This competition could decrease the enzyme’s ability to clear Aβ, potentially leading to the accumulation of amyloid plaques, which are a key feature of AD [50] (Figure 1). Indeed, some studies indicate that hyperinsulinemia might increase the risk of AD by hindering the clearance of Aβ, as IDE becomes overwhelmed by excessive insulin [51]. It is important to note that the competition hypothesis is a topic of debate. While IDE has a higher affinity for degrading insulin compared to Aβ, with Km values of approximately 100 μM for insulin and over 2 μM for Aβ, the physiological concentrations of insulin in the brain (around 3.8 pM) are significantly lower than IDE’s Km. Some researchers suggest that, under normal conditions, the significant competitive inhibition of Aβ degradation by insulin is questionable. As a result, it has been proposed by some researchers that the concept of direct competition should be reconsidered [20,52,53] (Figure 1).

There is compelling evidence that insulin resistance, a defining feature of T2DM, significantly elevates the risk of developing AD. In T2DM, the body becomes resistant to insulin, resulting in chronically high insulin levels. This state directly impairs the ability of the IDE to effectively clear Aβ from the brain. The connection between insulin resistance and AD has solidified the concept of “Type 3 Diabetes,” which posits that insulin resistance in the brain, combined with an impaired capacity to eliminate Aβ, plays a critical role in the onset of AD. If IDE prioritizes the clearance of insulin over Aβ, it will undoubtedly exacerbate the accumulation of amyloid plaques and accelerate the progression of neurodegenerative symptoms [54,55].

During the aging process, the expression of IDE in the brain declines, which directly impairs the clearance of Aβ and accelerates the development of AD [56]. Studies with animal models confirm that lower levels of IDE correlate strongly with increased formation of amyloid plaques. In contrast, evidence demonstrates that enhancing IDE activity effectively improves the clearance of Aβ, positioning IDE as a critical therapeutic target for AD [57].

## 3. Genetic and Structural Considerations

Recent studies have elucidated the genetic and structural dimensions of IDE and its implications in metabolic and neurodegenerative diseases.

Genetic polymorphisms in the IDE gene have been linked to an increased risk of both T2DM and AD, potentially due to reduced enzymatic efficiency [58]. For instance, certain single-nucleotide polymorphisms (SNPs) in the IDE gene have been associated with an elevated risk of late-onset AD, independent of the apolipoprotein E (APOE) ε4 allele [59].

Among the genetic variants investigated in relation to AD, the rs1887922 polymorphism within the *IDE* gene has garnered attention for its potential role in modulating disease risk. A recent meta-analysis, which included four independent studies, identified a significant association between rs1887922 and increased susceptibility to AD, with odds ratios greater than 1 across four genetic models [60]. Moreover, a case-control study involving 406 Han Chinese individuals from the Xinjiang region investigated the association between two IDE gene SNPs (rs1887922 and rs1999764) and LOAD risk. The CT + CC genotype of rs1999764 was found to exert a protective effect compared to the TT genotype (adjusted *p* = 0.0001; OR = 0.226), whereas the CT + CC genotype of rs1887922 was associated with a higher risk of LOAD (adjusted *p* = 0.0001; OR = 3.640). These associations were independent of the apolipoprotein E ε4 polymorphism and displayed sex-specific patterns, with rs1887922 being more relevant in men and rs1999764 in women [59].

Another study investigated the association between the rs2421943 SNP in the IDE gene and the risk of mild cognitive impairment (MCI) and AD. The study included two independent cohorts totaling 1670 individuals (595 AD patients, 400 controls; 135 MCI patients, 540 controls). Genotypic analysis by PCR and restriction enzyme digestion revealed that AG and GG genotypes were significantly associated with an increased risk of AD, while AG was also linked to a higher risk of MCI. In silico analysis showed that rs2421943 lies within a predicted binding site for hsa-miR-7110-5p, suggesting a possible post-transcriptional regulatory mechanism that may reduce IDE levels, impair Aβclearance, and thereby contribute to disease progression. These findings support rs2421943 as a genetic risk factor for both AD and its prodromal stage [61].

Additionally, bioinformatics analyses have identified shared genetic factors between T2DM and AD, with several SNPs common to both disorders, suggesting a potential overlap in their pathophysiological mechanisms [62].

Structurally, IDE possesses a flexible catalytic chamber capable of accommodating a range of substrates—including insulin, Aβ, and islet amyloid polypeptide (IAPP) [18,63]. This structural versatility, while functionally advantageous, presents challenges for the development of highly selective therapeutic modulators. Recent research [64] has highlighted the role of specific hydrophobic and aromatic residues within the IDE active site in substrate binding and catalysis, underscoring the complexity of targeting IDE for therapeutic purposes.

## 4. Preclinical Findings on the Relationship Among IDE, Insulin, and Aβ

A growing body of preclinical evidence highlights the pivotal role of IDE at the intersection of metabolic regulation and Aβ pathology, particularly within the context of T2DM and AD.

An initial study demonstrated that IDE activity and mRNA levels were significantly reduced in the cerebral cortex of STZ-induced diabetic rats, leading to increased Aβ deposition in brain tissues [65].

Similar results were observed in the GK diabetic rat model, where missense mutations of IDE impaired Aβ degradation and promoted the formation of Aβx-40 and Aβx-42 aggregates [66]. In AD transgenic mice, IDE concentrates at the edges of Aβ plaques in the cerebral cortex, where it probably undergoes an oxidation process that reduces its activity [18]. Consistently, in the same mouse models, IDE oxidation in the hippocampus is higher than in the cerebellum, thus initiating a vicious circle that further favors the formation and accumulation of Aβ and the increase in oxidative stress levels [18].

One of the first pieces of evidence concerns competition for substrates and the saturation of IDE. In IDE knockout mouse models, insulin degradation in liver and brain tissues is reduced by approximately 58–72%, resulting in chronic hyperinsulinemia and significantly elevated Aβ levels in the brain [67]. However, studies conducted on IDE knockout (IDE-KO) mice have provided conflicting results. In some mouse lines, marked glucose intolerance and hyperinsulinemia are observed, while in others, hyperinsulinemia does not appear [15].

Liver-specific IDE knockout (L-IDE-KO) mice exhibit insulin resistance and glucose intolerance despite normal insulin levels and preserved β-cell function. These effects are associated with reduced insulin receptor (IR) expression and signaling, including impaired phosphorylation of IR and AKT, and increased expression of gluconeogenic genes. Under high-fat diet, L-IDE-KO mice develop compensatory hyperinsulinemia, further supporting a key role for hepatic IDE in maintaining systemic insulin sensitivity through both enzymatic and non-enzymatic mechanisms [68,69].

Moreover, recent findings from β-cell-specific IDE knockout (B-IDE-KO) mouse models have shed light on the role of IDE in pancreatic β-cell function and maturation. Unlike total IDE knockout mice, which often show systemic metabolic disturbances, B-IDE-KO mice display only mild glucose intolerance, with no significant changes in fasting glucose levels. However, these mice exhibit elevated plasma C-peptide levels and constitutive insulin secretion, suggesting a dysregulation of glucose-stimulated insulin secretion (GSIS) [70]. This secretory phenotype is associated with decreased expression of GLUT2 and increased GLUT1 at the β-cell membrane, which may promote insulin release even at low glucose concentrations while impairing responsiveness to high glucose. These alterations in glucose transporter expression point toward a state of β-cell immaturity. Interestingly, these findings imply that IDE plays a critical role not only in insulin degradation but also in the developmental and functional maturation of β-cells, possibly through its impact on glucose sensing and granule processing. Moreover, increased expression of hepatic gluconeogenic genes in B-IDE-KO mice suggests that IDE might also influence pancreas–liver crosstalk, further highlighting its role in systemic glucose homeostasis [71].

Although substantial evidence supports a role for IDE in insulin degradation within cultured cells, its precise site of action remains unclear. IDE is predominantly localized in the cytosol and lacks a classical signal peptide, raising questions about how it accesses extracellular or endosomal insulin. Some studies, such as Zhao et al., suggest that IDE can be secreted via an unconventional pathway [72]. However, other reports, including that of Song et al., have challenged this notion, showing that IDE secretion from cells like HEK293 and BV2 occurs at extremely low levels, comparable to the nonspecific release of other cytosolic proteins such as lactate dehydrogenase [73]. Another key aspect still to be clarified is whether IDE-mediated Aβ degradation depends on the PI3K/Akt/GSK-3β pathway. Insulin signaling through the phosphoinositide 3-kinase (PI3K)/Akt pathway (Figure 2) has been shown to upregulate the expression of IDE in hippocampal neurons, by approximately 25%, suggesting a compensatory response to insulin resistance [74]. For instance, insulin treatment has been reported to increase the synthesis of postsynaptic density protein 95 (PSD-95), a marker of synaptic plasticity, via the PI3K-Akt-mammalian target of rapamycin (mTOR) signaling pathway in hippocampal area CA1 neurons [75].

In support of this, administration of bis ethyl maltolate oxidovanadate (BEOV) to APP transgenic mice significantly increased phosphorylated PI3K (p-PI3K) and Akt (p-Akt) levels while decreasing phosphorylated GSK-3β (p-GSK-3β) in the brain, leading to elevated IDE levels, reduced Aβ and p-tau deposits, and improved cognitive function [76].

In another study, AD was induced in adult albino mice by intracerebral injection of STZ, with cognitive impairment confirmed via the Morris water maze test. Treated mice exhibited significantly lower levels of p-PI3K, p-Akt, and p-GSK-3β in the hippocampus and higher levels of p-tau and Aβ1-42 compared to healthy controls. In contrast, pioglitazone and a kefir-derived probiotic reversed these alterations [77]. Akhtar et al. [78] also developed an AD model in Wistar rats via intracerebroventricular STZ injection, finding that oral administration of sodium orthovanadate (SOV) significantly improved behavioral test performance. On a molecular level, SOV activated the PI3K/Akt/GSK-3β pathway, leading to increased IDE levels and decreased p-tau and Aβ while upregulating insulin receptors (IR) and IRS-1 in the rat brain.

These results indicate that insulin can modulate protein synthesis in neuronal dendrites through the activation of the PI3K-Akt-mTOR pathway.

Some studies investigated the activation of microglia and astrocytes as another potential mechanism to enhance IDE secretion and facilitate the clearance of Aβ deposits [79,80]. Microglia, activated in neurodegenerative diseases, release vesicles containing IDE that degrade extracellular Aβ [19]. Additionally, glial cells can bind to and phagocytose Aβ, thereby eliminating it [35], but reactive astrocytes, generated in response to cellular damage or toxic molecules, lose this ability [36,37]. Despite these findings, there are conflicting data regarding glial cell activation; at the same time, some studies suggest that glial hyperactivation contributes to neuroinflammation, and others indicate that glial cells may promote Aβ clearance through IDE secretion or direct phagocytosis [81,82].

Corraliza-Gomez et al. (2023) [40] proposed that IDE modulates microglial phenotypes through non-enzymatic mechanisms, influencing key aspects such as cytokine production, myelin phagocytosis, inflammatory responses, and cell proliferation. Their findings suggest that the absence of IDE significantly alters microglial plasticity, not by impairing its proteolytic activity on substrates like Aβor insulin, but rather through a regulatory role that is independent of its enzymatic function.

Other studies have suggested that IDE may contribute to the modulation of microglial activation states, promoting a shift toward an anti-inflammatory phenotype. This effect appears to be mediated not through its enzymatic degradation of Aβor insulin, but via non-catalytic mechanisms that influence microglial responses to stress and cytokine signaling [83,84].

Recent studies have explored the consequences of IDE deletion in brain regions such as the hippocampus and olfactory bulb—areas involved in memory and early AD pathology. In IDE knockout mice, selective microglial activation was observed in the hippocampus without associated changes in volume or astrocyte activation, indicating a targeted neuroimmune response. Additionally, memory performance in older mice appeared to correlate with IDE gene dosage. In the olfactory bulb, where insulin signaling is particularly active, IDE deficiency produced distinct molecular and behavioral alterations, defining a specific phenotypic signature. In vitro, IDE-deficient microglia showed impaired adaptability to environmental cues and altered phenotypic plasticity, with only transient changes in their ability to handle Aβ. These findings point toward a novel, non-proteolytic role for IDE in regulating microglial function, possibly through mechanisms related to cellular localization and signaling integration rather than direct peptide degradation [40].

Other studies on animal models have investigated compensatory mechanisms and tissue-specific dynamics. Indeed, although IDE predominantly degrades soluble forms of Aβ, other enzymes such as neprilysin (NEP) compensate by targeting insoluble aggregates [85]. NEP is particularly efficient at degrading insoluble, aggregated, and fibrillar forms of Aβ, as well as shorter cleavage products (e.g., Aβ16), playing a central role in mitigating plaque accumulation and modulating Aβ aggregation kinetics. In contrast, IDE preferentially degrades soluble, monomeric Aβ forms (primarily Aβ40) and has limited ability to process peptide fragments or aggregated Aβ. Knockout or deficiency models further support this dichotomy: NEP deficiency in mice leads to a pronounced increase in plaque burden, while IDE loss primarily elevates soluble Aβ levels. Thus, NEP appears to be the primary regulator of extracellular Aβ deposition, whereas IDE chiefly governs the clearance of soluble Aβ species [36,86].

Furthermore, IDE activity exhibits tissue-specific variability, being most prominent in insulin-sensitive organs such as the liver and pancreas. However, brain-specific deficits in IDE function have direct consequences for Aβ pathology and cognitive outcomes [20].

## 5. Clinical Evidence on the Relationship Between IDE, Insulin, and Aβ

Numerous clinical and epidemiological studies have established a strong connection between T2DM and an increased risk of developing AD, particularly in individuals who do not carry the APOE ε4 allele [87]. For instance, Marseglia et al. reported that T2DM increased dementia risk predominantly in individuals without the APOE ε4 allele, suggesting that diabetes-related mechanisms may contribute independently to AD pathology in this group. Conversely, other studies have shown that APOE ε4 carriers with T2DM may have an even higher risk, reflecting complex gene–environment interactions [87]. As is known, hyperinsulinemia critically links metabolic dysfunction to neurodegeneration. It saturates IDE, impairing the clearance of Aβ and promoting its accumulation in the brain. As IDE degrades both insulin and Aβ, it serves as a vital connection between these processes [88].

Postmortem analyses of AD patients have revealed decreased IDE expression in brain regions with heavy Aβ plaque deposition, and this reduction has been correlated with disease severity [89]. In patients with T2DM, IDE function appears to be disrupted in a tissue-specific manner—such as reduced activity in the liver and muscle—while paradoxically, circulating levels of IDE in the serum may be elevated, reflecting complex systemic dysregulation [5].

Several clinical studies have supported the systemic dysregulation of IDE in the context of T2DM and cognitive impairment. For example, Farris et al. found that IDE protein levels were significantly reduced in the hippocampus of AD patients compared to controls, with levels inversely correlating with Aβ accumulation [14].

A study investigating serum IDE levels across 120 individuals—including patients with type 2 diabetes mellitus (T2DM), Alzheimer’s disease (AD), and healthy controls—found that IDE levels were significantly elevated in T2DM patients compared to the other groups, after adjusting for age and sex. IDE correlated positively with metabolic markers such as BMI, fasting glucose, C-peptide, HbA1c, insulin resistance, and triglycerides. Interestingly, stratified analyses revealed a negative partial correlation between IDE and HbA1c in AD patients, and between IDE and C-peptide in healthy individuals, while no significant correlations were observed in T2DM patients. These findings suggest that circulating IDE levels are linked to metabolic function and may reflect metabolic status, although IDE is not currently applicable as a clinical biomarker [90].

An observational study on 146 patients with type 2 diabetes found that serum IDE levels were higher in those with mild MCI compared to controls. However, within the MCI group, higher IDE levels correlated with better cognitive performance (MoCA scores). Overall, IDE was positively associated with cognition and negatively with insulin resistance and glycemic parameters. Logistic regression confirmed IDE as an independent factor associated with MCI. These findings suggest that lower IDE levels may contribute to cognitive decline in diabetic patients [80]. Because the major hypothesis suggests that in chronic hyperinsulinemia, insulin competes with Aβ for IDE binding, impairing Aβ clearance and promoting its accumulation in the brain, although this mechanism is well supported by preclinical data, it remains difficult to quantify directly in human studies.

Nevertheless, clinical evidence has shown that individuals with T2DM exhibit higher cerebral levels of Aβ, consistent with impaired IDE-mediated clearance [91]. A groundbreaking study analyzing data from the Alzheimer’s Disease Neuroimaging Initiative reveals a striking contrast in cerebrospinal fluid (CSF) Aβ1-42 levels between individuals with T2DM and non-diabetic controls. The findings show that T2DM patients have significantly elevated levels of CSF Aβ1-42, raising compelling questions about the underlying mechanisms—whether this reflects an increase in soluble Aβ within the brain’s interstitial fluid or an impairment in the brain’s ability to clear it.

Even more intriguing, these individuals displayed reduced levels of cerebral cortical Aβ deposition when assessed through PET imaging in key brain regions such as the anterior cingulate, precuneus, and temporal lobe. This unexpected inverse relationship between CSF Aβ1-42 and cortical Aβ deposition is not only consistent across various diabetes statuses but is especially pronounced in T2DM subjects [91].

The formation of Aβ is a hallmark of Alzheimer’s and contributes significantly to the disease’s progression. The elevated levels of Aβ1-42 in CSF observed in people with T2DM highlight a critical issue: impaired IDE activity. IDE plays a vital role in degrading both insulin and Aβ, and in cases of hyperinsulinemia common in T2DM, the enzyme can become overwhelmed. This saturation leads to a significant reduction in Aβ clearance, consequently raising soluble Aβ levels in the CSF [91].

For easier understanding and comparison of the studies mentioned in this section, they are presented in Table 1.

However, due to technical and ethical limitations, most clinical studies have been limited to phenotypic assessment by serum or plasma analysis without investigating the underlying mechanisms.

Most studies investigating pharmacological therapies for diabetic cognitive impairment have focused on changes in blood glucose levels and IDE expression within the CNS, leaving the impact on peripheral IDE levels and glucose regulation mechanisms less understood.

## 6. Therapeutic Perspectives: Modular IDE to Counteract AD

Several clinical protocols dedicated to the testing of Alzheimer’s disease-modifying drugs are underway to continue the fight against this disease.

Although modulating IDE represents an innovative strategy, challenges remain related to the intervention specificity and the systemic effects of regulating the enzyme.

Researchers are actively pursuing innovative approaches, including the development of compounds that either activate IDE or inhibit its insulin-binding capacity to prioritize Aβ clearance.

For example, a synthetic peptide derived from the pregnancy-associated immunomodulatory factor (PIF) has been shown to compete with insulin and Aβ for IDE binding, resulting in decreased Aβ levels in APP-transfected neuronal cells. The effect is abolished by IDE inhibitors such as N-ethylmaleimide (NEM), further confirming IDE’s role in Aβ degradation [92]. Finally, interventions targeting IDE must be implemented with caution, as any disruption of insulin clearance could lead to exacerbations in hyperinsulinemia and worsen insulin resistance [93].

Moreover, a peptide derived from varicella-zoster virus (VZV) has been engineered to selectively inhibit IDE without interacting with its catalytic zinc ion, thereby minimizing off-target toxicity. This peptide binds specifically within the central cavity of IDE and effectively suppresses its activity in vitro. In mouse models of type 1 and type 2 diabetes, it ameliorated insulin-related defects and appeared to modulate the pro-inflammatory profile of insulin-reactive CD4+ T cells. These findings support its potential as a targeted and safe therapeutic strategy for diabetes [23].

One of the most significant advances has been the identification of allosteric activation sites within IDE’s structure. High-resolution structural analyses revealed that IDE can exist in “open” (active) and “closed” (inactive) conformations. Small molecules—particularly indole-based compounds and nucleotide-mimetic activators—have been shown to bind these sites, significantly increasing IDE’s catalytic activity toward Aβ. These findings provide the foundation for the development of brain-penetrant drugs that boost IDE function, specifically in the central nervous system. Indeed, indole-based pharmacological tools have been optimized to simultaneously enhance Aβ degradation and preserve insulin clearance, a crucial step for the safety profile of IDE-targeted therapies [24].

In addition to allosteric modulation, chemical strategies have emerged to improve IDE’s substrate specificity. Recent investigations have targeted tyrosine residues within IDE’s active site that are involved in substrate recognition. By chemically modifying these residues, researchers have succeeded in shifting IDE’s activity to preferentially degrade Aβ over insulin [25].

Future research should clarify the interaction among IDE, blood glucose, and Aβ in both brain and blood to better identify the mechanisms underlying diabetes-associated cognitive decline. Furthermore, the possible use of IDE as a biomarker of Aβ deposition, Aβ clearance, early diagnosis, and progression of diabetes-related cognitive impairment should also be explored [80].

While a growing number of studies investigate the etiological factors of AD beyond Aβ, this peptide continues to represent the principal therapeutic target.

It is no coincidence that since 2021, several anti-amyloid antibodies have been approved in the USA and are in the approval phase in Europe for the treatment of the disease (such as Lecanemab and Donanemab) that have proven to be effective in removing brain amyloid and significantly slowing the progression of cognitive symptoms [94].

Anyway, research continues actively to develop next-generation treatments targeting amyloid, tau pathology, neuroinflammation, and other mechanisms [95]. However, modulating IDE activity is not just an area of interest; it is a promising treatment strategy for both diabetes and AD. Increasing IDE activity can significantly enhance insulin sensitivity in diabetes while simultaneously facilitating the removal of Aβ in AD [50].

In the future, it will be essential to develop IDE modulators with brain specificity and minimal systemic effects. Targeted delivery strategies, such as nanoparticles or exosomes, could enhance selectivity within the CNS. Multi-omics approaches may help clarify IDE regulatory mechanisms under pathological conditions. Additionally, longitudinal studies in diabetic patients with MCI could define its role as a biomarker. Combined approaches with anti-tau or anti-inflammatory therapies may further enhance clinical efficacy. Finally, identifying IDE genetic variants predictive of therapeutic response could support the development of personalized treatments.

## 7. Conclusions

Multiple factors can alter the expression or activity of IDE, including neuroinflammation, high-fat diets, genetic polymorphisms, oxidative stress, and brain aging (Figure 3). Among these, hyperglycemia and insulin resistance emerge as central contributors to cognitive impairment associated with both diabetes mellitus and AD, two conditions that share the pathological hallmark of Aβ deposition in the brain. Insulin resistance not only exacerbates hyperinsulinemia but also promotes Aβ accumulation, due in part to the competitive role of IDE, which degrades both insulin and Aβ. In hyperinsulinemic states, IDE preferentially targets insulin, thereby diminishing its capacity to degrade Aβ. This mechanism may further perpetuate insulin resistance and Aβ accumulation, ultimately accelerating cognitive decline (Figure 3).

While cell culture studies provide valuable mechanistic insights into IDE function and its interplay with metabolic and neurodegenerative pathways, these findings should be interpreted as preliminary, proof-of-concept evidence. They offer a foundation for understanding disease mechanisms but do not yet support clinical application. Further in vivo validation and rigorous assessment of safety and efficacy are necessary before translating these findings into therapeutic strategies.

Enhancing IDE-mediated Aβ degradation represents a promising therapeutic approach in AD; however, such strategies must be approached with caution due to potential systemic effects. Given that insulin is a primary substrate of IDE, therapeutic enhancement of IDE activity could impair insulin clearance, worsening hyperinsulinemia and insulin resistance—especially in elderly or metabolically vulnerable individuals. This underscores the importance of maintaining a delicate balance between promoting Aβ clearance and preserving systemic metabolic homeostasis. To mitigate these risks, future approaches may involve selectively modulating IDE activity within the central nervous system, utilizing transient or context-dependent activation, or designing targeted molecules that enhance IDE’s affinity for Aβ while minimizing effects on insulin degradation.

## Figures and Tables

**Figure 1 ijms-26-06693-f001:**
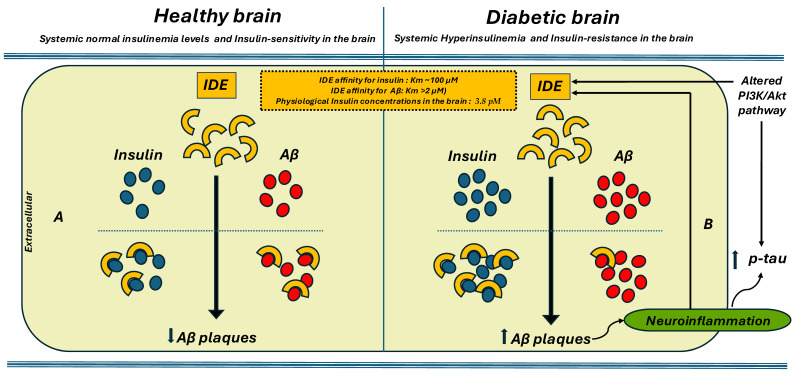
**Insulin-degrading enzyme (IDE) activity and Aβ degradation.** IDE plays a crucial role in maintaining the balance between the production and degradation o Aβ peptides in the healthy brain (**A**). However, in cases of hyperinsulinemia, reduced IDE activity impairs the clearance of Aβ, leading to its accumulation in the brain and an increase in Aβ plaques (**B**). The competition hypothesis is debated. Although IDE has a higher affinity for insulin (Km~100 μM) than for Aβ (Km > 2 μM), the brain’s insulin levels (~3.8 pM) are far below this threshold. In diabetic patients, the accumulation of Aβ plaques leads to increased neuroinflammation, which in turn contributes to higher tau protein levels. This increase in phosphorylated tau (p-tau) is further driven by alterations in the PI3K-Akt pathway.

**Figure 2 ijms-26-06693-f002:**
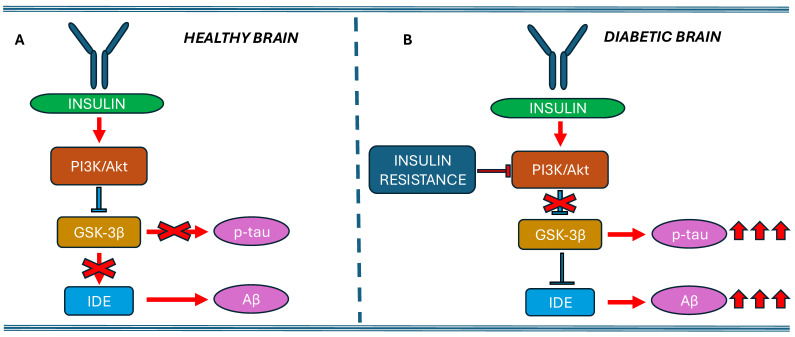
**Potential link between insulin resistance, PI3K/Akt signaling, GSK-3β modulation, and IDE activity.** Under normal conditions (insulin sensitivity), insulin binds to its receptor, leading to the activation of PI3K and subsequently Akt. Akt phosphorylates and inhibits GSK-3β, keeping it inactive. This regulation helps control tau phosphorylation, promotes the activity of the insulin-degrading enzyme (IDE), and prevents the accumulation of Aβ (**A**). Under insulin-resistant conditions, the insulin receptor becomes less responsive, resulting in reduced activation of the PI3K/Akt signaling pathway. Consequently, Akt fails to inhibit GSK-3β, which remains active. The persistent activation of GSK-3β contributes to the hyperphosphorylation of tau, inhibition of insulin-degrading enzyme (IDE) activity, and increased production and accumulation of Aβ, all of which are key pathological features of Alzheimer’s disease (**B**). *Red crosses*: indicate the inhibition of that pathway or its reduction.

**Figure 3 ijms-26-06693-f003:**
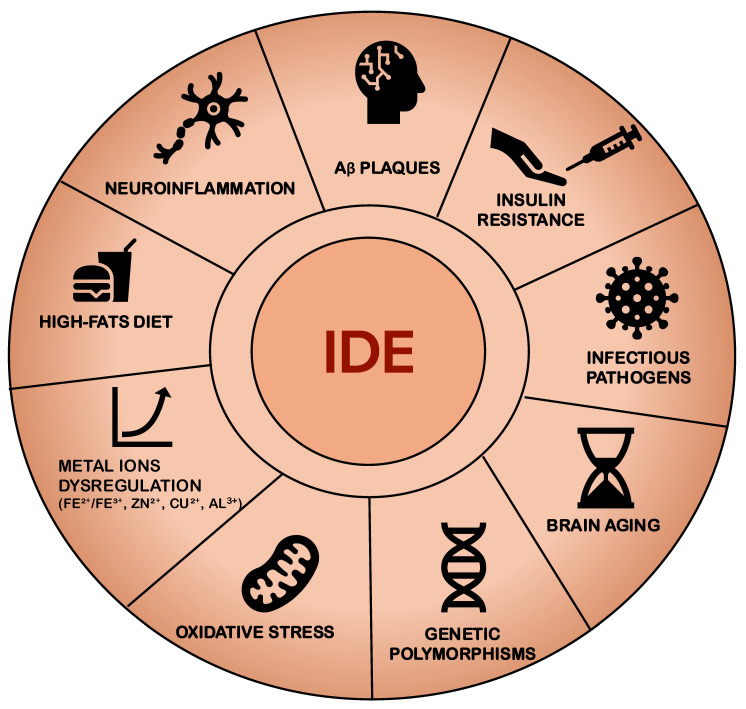
**Pathophysiological and environmental factors influencing IDE activity.** Multiple elements—including neuroinflammation, Aβ plaque accumulation, insulin resistance, infectious pathogens, brain aging, genetic polymorphisms, oxidative stress, dysregulation of metal ions (Fe^2+^/Fe^3+^, Zn^2+^, Cu^2+^, Al^3+^), and high-fat diet—contribute to reduced IDE expression or enzymatic activity. This impairment hinders the degradation of insulin and Aβ, promoting neurodegenerative and metabolic processes characteristic of T2DM and Alzheimer’s disease [25,40,54,90,96].

**Table 1 ijms-26-06693-t001:** Overview of the studies referenced in the text, illustrating the interplay among IDE, insulin, and Aβ.

Study	Population	Biomarkers Assessed	Aβ Burden	Results
[74]	146 patients with T2DM, with and without MCI	Serum IDE, blood glucose, HOMA-IR, HOMA-IR	Not assessed	IDE was an independent predictor of MCI; higher levels associated with better cognitive scores
[81]	2305 cognitively intact individuals aged ≥ 60 years	Neuropsychological performance	Not assessed	Uncontrolled T2DM linked to deficits in perceptual speed, fluency, and primary memory in APOEε4 non-carriers
[83]	Post-mortem human AD brains	Regional brain proteomics	Indirectly assessed via related proteins	Decreased IDE expression in brain regions with heavy Aβ plaque deposition
[85]	ADNI participants with and without T2DM	CSF Aβ1-42, cortical Aβ (PET)	↑ CSF Aβ41-42 in T2DM; ↓ cortical Aβ	T2DM associated with ↑ CSF Aβ1-42 but ↓ cortical Aβ in specific regions

T2DM: Type 2 Diabetes Mellitus; MCI: Mild Cognitive Impairment; AD: Alzheimer’s disease; ADNI: Alzheimer’s Disease Neuroimaging Initiative; HOMA-IR: Homeostatic Model Assessment–Insulin Resistance; CSF: Cerebrospinal fluid; PET: Positron Emission Tomography; ↑: Increase; ↓: Decrease.

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
