# Peer review of "Understanding the Insulin-Degrading Enzyme: A New Look at Alzheimer’s Disease and Aβ Plaque Management"

_ijms, 2025, doi:10.3390/ijms26146693_

Round 1
Reviewer 1 Report
Comments and Suggestions for Authors
- In conditions such as type 2 diabetes (T2DM), where the body's cells become less responsive to 53 insulin—known as insulin resistance—this regulation is impaired, leading to hyperinsulinemia [2]. 54 When IDE function is compromised, excess insulin can accumulate in the bloodstream, worsening 55 insulin resistance, disrupting glucose metabolism, and contributing to the progression of 56 diabetes[3]. Cite relevant studies https://doi.org/10.1016/j.ijbiomac.2022.03.004
- IDE is particularly important in managing Aβ accumulation in Alzheimer's disease (AD) and 62 diabetic cognitive impairment. By increasing IDE levels in the central nervous system (CNS), we 63 could improve the breakdown of Aβ plaques, potentially leading to significant cognitive 64 enhancements[8]. https://doi.org/10.1021/acsomega.2c06634
- For example, administration of 6bK to subjects with T2DM led….6Bk?
- Beta-amyloid (Aβ) is a peptide or Amloid beta, maintain consistency.
- https://doi.org/10.3390/molecules27144652; This study provides important insights into the connection of diabetes and AD.
- Figure 1 needs improvement, It is oversimplified.
- Another key aspect still to be clarified is whether IDE-mediated Aβ degradation depends on the 205 PI3K/Akt/GSK-3β pathway. Insulin signaling through the phosphoinositide 3-kinase (PI3K)/Akt 206 pathway has been shown to upregulate the expression of IDE in hippocampal neurons, by approx- 207 imately 25%, suggesting a compensatory response to insulin resistance. strongly recommend the inclusion of a schematic figure illustrating the relevant signaling pathways. A visual representation would significantly enhance the clarity of the mechanistic relationships described in the text, particularly the potential link between insulin resistance, PI3K/Akt signaling, GSK-3β modulation, and IDE activity.
- Section 5, CLINICAL EVIDENCE ON THE RELATIONSHIP BETWEEN IDE, INSULIN, AND Aβ . Presenting key findings from relevant clinical studies—including patient populations, biomarkers assessed, IDE expression levels, insulin signaling status, and Aβ burden—would provide a clearer and more accessible overview of the current evidence base. Such a table would enhance the reader's understanding and allow for easier comparison across studies.
- THERAPEUTIC PERSPECTIVES: MODULAR IDE TO COUNTERACT AD, recent updates in the therapeutic targeting must be incorporated.
- Figure 2. Factors that alter the expression or activity of the Insulin-Degrading Enzyme (IDE) . Too simple and must be updated.
- Future perspective in this domain of research along with recent updates must be added.
Author Response
Reviewer 1
- In conditions such as type 2 diabetes (T2DM), where the body's cells become less responsive to insulin—known as insulin resistance—this regulation is impaired, leading to hyperinsulinemia [2]. When IDE function is compromised, excess insulin can accumulate in the bloodstream, worsening insulin resistance, disrupting glucose metabolism, and contributing to the progression of diabetes [3]. Cite relevant studieshttps://doi.org/10.1016/j.ijbiomac.2022.03.004
We thank the reviewer for this important suggestion. As requested by the reviewer, we have updated the recommended bibliography (Ref 2: Shahwan M, Alhumaydhi F, Ashraf GM, Hasan PMZ, Shamsi A. Role of polyphenols in combating Type 2 Diabetes and insulin resistance. Int J Biol Macromol. 2022 May 1;206:567-579. doi: 10.1016/j.ijbiomac.2022.03.004) and, accordingly, we considered it appropriate to add the following sentence (see lines 50-53): "In order to improve glucose uptake, numerous studies have focused on elucidating the underlying mechanisms of insulin resistance as well as exploring potential therapeutic interventions, both pharmacological and non-pharmacological."
- IDE is particularly important in managing Aβ accumulation in Alzheimer's disease (AD) and diabetic cognitive impairment. By increasing IDE levels in the central nervous system (CNS), we could improve the breakdown of Aβ plaques, potentially leading to significant cognitive enhancements[8]. https://doi.org/10.1021/acsomega.2c06634
We thank the reviewer for this important suggestion. As requested by the reviewer, we have updated the recommended bibliography (Ref 3: Atiya A, Das Gupta D, Alsayari A, Alrouji M, Alotaibi A, Sharaf SE, Abdulmonem WA, Alorfi NM, Abdullah KM, Shamsi A. Linagliptin and Empagliflozin Inhibit Microtubule Affinity Regulatory Kinase 4: Repurposing Anti-Diabetic Drugs in Neurodegenerative Disorders Using In Silico and In Vitro Approaches. ACS Omega. 2023 Feb 7;8(7):6423-6430. doi: 10.1021/acsomega.2c06634), andaccordingly, we considered it appropriate to add the following sentence (see lines 53-57): “Moreover, numerous studies have suggested a potential shared pathophysiology between Type 2 Diabetes and Alzheimer’s disease. As a result, there is growing interest in recent years in research aimed at deciphering the interplay mechanisms between the two conditions, including the possible role of antidiabetic drugs and their therapeutic potential in Alzheimer’s disease and related neurodegenerative disorders.”
- For example, administration of 6bK to subjects with T2DM led….6Bk?
We confirm that the correct notation is 6bK, not 6Bk.
- Beta-amyloid (Aβ) is a peptide or Amloid beta, maintain consistency.
We apologize for the error. We have replaced "Beta-amyloid" with "Amyloid-beta."
- https://doi.org/10.3390/molecules27144652; This study provides important insights into the connection of diabetes and AD.
We thank the reviewer for this important suggestion. As requested by the reviewer, we have updated the recommended bibliography (Ref 4: Ashraf, G. M., DasGupta, D., Alam, M. Z., Baeesa, S. S., Alghamdi, B. S., Anwar, F., Alqurashi, T. M. A., Sharaf, S. E., Al Abdulmonem, W., Alyousef, M. A., Alhumaydhi, F. A., & Shamsi, A. (2022). Inhibition of Microtubule Affinity Regulating Kinase 4 by Metformin: Exploring the Neuroprotective Potential of Antidiabetic Drug through Spectroscopic and Computational Approaches. Molecules, 27(14), 4652. https://doi.org/10.3390/molecules27144652), and accordingly, we considered it appropriate to add the following sentence (see lines 58-61): “In particular, scientific studies have highlighted how metformin can be used as part of a treatment strategy for diabetes associated with neurodegenerative diseases since it is an inhibitor of Microtubule affinity regulating kinase 4 (MARK4). MARK4 plays a key role in tau phosphorylation by targeting the Ser262 residue within the KXGS motif, leading to tau detachment from microtubules”.
- Figure 1 needs improvement, It is oversimplified.
Thank you very much for the suggestion. We have improved Figure 1 by adding p-Tau, neuroinflammation, and the altered PI3K/Akt pathway.
- Another key aspect still to be clarified is whether IDE-mediated Aβ degradation depends on the 205 PI3K/Akt/GSK-3β pathway. Insulin signaling through the phosphoinositide 3-kinase (PI3K)/Akt 206 pathway has been shown to upregulate the expression of IDE in hippocampal neurons, by approximately 25%, suggesting a compensatory response to insulin resistance. strongly recommend the inclusion of a schematic figure illustrating the relevant signaling pathways. A visual representation would significantly enhance the clarity of the mechanistic relationships described in the text, particularly the potential link between insulin resistance, PI3K/Akt signaling, GSK-3β modulation, and IDE activity.
Thank you for the valuable suggestion. We have created a visual representation (Figure 2) to illustrate the mechanistic relationships described in the text, particularly the potential link between insulin resistance, PI3K/Akt signaling, GSK-3β modulation, and IDE activity (see lines 313-329).
- Section 5, CLINICAL EVIDENCE ON THE RELATIONSHIP BETWEEN IDE, INSULIN, AND Aβ. Presenting key findings from relevant clinical studies—including patient populations, biomarkers assessed, IDE expression levels, insulin signaling status, and Aβ burden—would provide a clearer and more accessible overview of the current evidence base. Such a table would enhance the reader's understanding and allow for easier comparison across studies.
Thank you for the suggestion. We have created (Table 1) summarizing the main information from the clinical studies mentioned in the paragraph (see line 457).
- THERAPEUTIC PERSPECTIVES: MODULAR IDE TO COUNTERACT AD, recent updates in the therapeutic targeting must be incorporated.
As required, we have added therapeutic approaches targeting IDE modulation as a strategy to counteract AD. Consequently, the section titled “Therapeutic Perspectives: Modulating IDE to Counteract AD” has been revised as detailed below. Specifically, the portions highlighted in green indicate previously existing text that has been repositioned, while the portions highlighted in yellow correspond to newly added content, as per the reviewer’s suggestions. In addition, three new references have been included to support the newly added content Ref. 23-24-25 (See lines 494-502-506).
- THERAPEUTIC PERSPECTIVES: MODULAR IDE TO COUNTERACT AD
Several clinical protocols dedicated to the testing of Alzheimer's disease-modifying drugs are underway, to continue the fight against this disease.
Most studies investigating pharmacological therapies for diabetic cognitive impairment have focused on changes in blood glucose levels and IDE expression within the CNS, leaving the impact on peripheral IDE levels and glucose regulation mechanisms less understood[87].
Although modulating IDE represents an innovative strategy, challenges remain related to the intervention specificity and the systemic effects of regulating the enzyme.
Researchers are actively pursuing innovative approaches, including the development of compounds that either activate IDE or inhibit its insulin-binding capacity to prioritize Aβ clearance.
For example, a synthetic peptide derived from the pregnancy-associated immunomodulatory factor (PIF) has been shown to compete with insulin and Aβ for IDE binding, resulting in decreased Aβ levels in APP-transfected neuronal cells. The effect is abolished by IDE inhibitors such as N-ethylmaleimide (NEM), further confirming IDE’s role in Aβ degradation[88]. Finally, interventions targeting IDE must be implemented with caution, as any disruption of insulin clearance could lead to exacerbations in hyperinsulinemia and worsen insulin resistance[89].
Moreover a peptide derived from varicella-zoster virus (VZV) has been engineered to selectively inhibit IDE without interacting with its catalytic zinc ion, thereby minimizing off-target toxicity. This peptide binds specifically within the central cavity of IDE and effectively suppresses its activity in vitro. In mouse models of type 1 and type 2 diabetes, it ameliorated insulin-related defects and appeared to modulate the pro-inflammatory profile of insulin-reactive CD4+ T cells. These findings support its potential as a targeted and safe therapeutic strategy for diabetes[23].
One of the most significant advances has been the identification of allosteric activation sites within IDE's structure. High-resolution structural analyses revealed that IDE can exist in “open” (active) and “closed” (inactive) conformations. Small molecules—particularly indole-based compounds and nucleotide-mimetic activators—have been shown to bind these sites, significantly increasing IDE’s catalytic activity toward Aβ. These findings provide the foundation for the development of brain-penetrant drugs that boost IDE function specifically in the central nervous system. Indeed, indole-based pharmacological tools have been optimized to simultaneously enhance Aβ degradation and preserve insulin clearance, a crucial step for the safety profile of IDE-targeted therapies[24].
In addition to allosteric modulation, chemical strategies have emerged to improve IDE's substrate specificity. Recent investigations have targeted tyrosine residues within IDE’s active site that are involved in substrate recognition. By chemically modifying these residues, researchers have succeeded in shifting IDE’s activity to preferentially degrade Aβover insulin[25].
Future research should clarify the interaction between IDE, blood glucose, and Aβ in both brain and blood, to better identify the mechanisms underlying diabetes-associated cognitive decline. Furthermore, the possible use of IDE as a biomarker of Aβ deposition, Aβ clearance, early diagnosis, and progression of diabetes-related cognitive impairment should also be explored[75].
While a growing number of studies investigate the etiological factors of AD beyond Aβ, this peptide continues to represent the principal therapeutic target.
It is no coincidence that since 2021, several anti-amyloid antibodies have been approved in the USA and are in the approval phase in Europe for the treatment of the disease (such as Lecanemab and Donanemab) that have proven to be effective in removing brain amyloid and significantly slowing the progression of cognitive symptoms[90]. Anyway, research continues actively to develop next-generation treatments targeting amyloid, tau pathology, neuroinflammation, and other mechanisms[91]. However, modulating IDE activity is not just an area of interest, but it is a promising treatment strategy for both diabetes and AD. Increasing IDE activity can significantly enhance insulin sensitivity in diabetes while simultaneously facilitating the removal of Aβ in AD[47].
In the future, it will be essential to develop IDE modulators with brain specificity and minimal systemic effects. Targeted delivery strategies, such as nanoparticles or exosomes, could enhance selectivity within the CNS. Multi-omics approaches may help clarify IDE regulatory mechanisms under pathological conditions. Additionally, longitudinal studies in diabetic patients with MCI could define its role as a biomarker. Combined approaches with anti-tau or anti-inflammatory therapies may further enhance clinical efficacy. Finally, identifying IDE genetic variants predictive of therapeutic response could support the development of personalized treatments.
- Figure 2. Factors that alter the expression or activity of the Insulin-Degrading Enzyme (IDE). Too simple and must be updated.
We improved the figure 2 (now figure 3) by adding additional factors. We have incorporated two additional factors that alter insulin-degrading enzyme (IDE) activity into the diagram. We also improved the caption and added relevant bibliographic references, as recommended by the other reviewer.
- Future perspective in this domain of research along with recent updates must be added.
Thank you for the suggestion, we have incorporated “the future perspectives” into
Chapter 6. The text is reported below (Lines 522-528).
“In the future, it will be essential to develop IDE modulators with brain specificity and minimal systemic effects. Targeted delivery strategies, such as nanoparticles or exosomes, could enhance selectivity within the CNS. Multi-omics approaches may help clarify IDE regulatory mechanisms under pathological conditions. Additionally, longitudinal studies in diabetic patients with MCI could define its role as a biomarker. Combined approaches with anti-tau or anti-inflammatory therapies may further enhance clinical efficacy. Finally, identifying IDE genetic variants predictive of therapeutic response could support the development of personalized treatments.”

Reviewer 2 Report
Comments and Suggestions for Authors
Cerasuolo et al. have reviewed IDE’s role in regulating insulin and degrading Aβ, highlighting its relevance in T2DM and AD and proposing its dual role as a therapeutic target. While the review offers a general overview, several key issues limit its quality. The claim that IDE degrades Aβ plaques is misleading, as IDE’s activity has only been shown against soluble Aβ monomers in vitro, with its in vivo role still debated. The manuscript lacks critical evaluation of controversial hypotheses, such as the “competition hypothesis” between insulin and Aβ for IDE, which lacks physiological relevance given brain insulin levels and substrate affinities. It also relies heavily on secondary sources, neglecting important primary studies, including genetic and in vivo research crucial to understanding IDE in AD and diabetes. Key concepts like IDE’s subcellular localization, presence in extracellular vesicles, and pH-dependent activity are missing or superficially addressed, reducing the manuscript’s depth and translational value. Overall, important mechanistic and genetic studies are insufficiently discussed or omitted, leading to an incomplete and oversimplified perspective.
I thank the authors for their efforts but note that significant issues must be addressed before publication. If these are comprehensively corrected—by revising overstatements, citing primary literature, including missing studies, and offering a balanced critical analysis—I recommend major revision. Otherwise, the manuscript is unsuitable for publication in its current form.
COMMENTS
Abstract
- The affirmation that IDE can break amyloid beta plaques is not correct.
- The seminal paper about IDE’s ability to degrade amyloid-ß peptides (Kurochkin and Goto, 1994; 1016/0014-5793(94)00387-4) was performed in vitro and assessed the degradation of synthetic monomeric Aß peptides Aß1-40 and Aß1-28.
- In Farris et al, 2003 (10.1073/pnas.0230450100), authors demonstrated Aß1-40 degradation deficits in soluble and brain membrane fractions. The work by Miller et al 2003 (10.1073/pnas.1031520100) reported increased Aß40 and Aß42 peptides accumulation in the brains from IDE deficient mice in an IDE dose-dependent manner.
- A very recent paper by Morito et al, 2025 (10.1523/JNEUROSCI.2152-24.2025) reports that neprilysin deficiency accelerates Aß plaque formation more prominently than IDE deficiency.
- The role of IDE in degrading Aß in vivo is questionable, as discussed by Corraliza-Gomez et al, 2023 (10.1186/s12974-023-02914-7). As discussed in that paper, a major question that needs to be addressed is where can IDE interact with Aß, since the protease and its putative substrate are localized in different subcellular compartments.
- The “competition hypothesis” between insulin and Aß for IDE’s protease activity is questionable. IDE degrades insulin with higher affinity than Aß (Km ~100 nM vs >2 μM, respectively (Hoyer, 2006), but insulin levels in the brain are ~3.8 pM (Geijselaers et al., 2017; 10.3233/JAD-170522), much lower than the insulin Km, and thus unlikely to competitively inhibit Aβ degradation. Therefore, it was suggested already in 2016 that the hypothesis about competitive inhibition of IDE needs to be reformulated (Pivovarova et al., 2016; 10.1080/07853890.2016.1197416).
Introduction
- It is worth noting that the reported substrates degraded by IDE have been primarily identified in vitro using biochemical or cell-free systems, which may not fully reflect physiological relevance (Duckworth et al., 1998; doi:10.1210/edrv.19.5.0349; Shen et al., 2006; doi:10.1038/nature05143)
- Regarding the last sentence of the introduction, although IDE has been implicated in Aβ catabolism, the notion that increasing IDE levels in the CNS would straightforwardly improve Aβ clearance and cognitive outcomes is oversimplified and not fully supported by in vivo evidence. In fact, IDE degrades a wide variety of substrates, including insulin, amylin, and other bioactive peptides, often with higher affinity than for Aβ (Farris et al., 2003; doi.org/10.1073/pnas.0230450100; Gonzalez-Casimiro et al., 2021; 10.3390/biomedicines9010086). This substrate promiscuity raises concerns that increasing IDE activity might disrupt other regulatory pathways, particularly those involved in insulin signaling.
IDE activity
- While the structural description of IDE is accurate, the authors may consider briefly expanding on its enzymatic activity and regulatory potential to provide a more complete overview. IDE is a zinc-dependent metalloprotease with broad substrate specificity, playing key roles in the degradation of insulin, amyloid-β, and other bioactive peptides. Its activity is modulated by substrate conformation, subcellular localization, and interactions with small-molecule effectors. Recent studies have identified several pharmacological modulators of IDE, including both inhibitors and activators, although their substrate selectivity and therapeutic applicability remain under investigation. Including a concise mention of these aspects would better contextualize the significance of IDE’s structure-function relationship.
- The authors note that IDE is evolutionarily conserved from bacteria to eukaryotes, implying functional preservation. I recommend expanding this point with reference to Corraliza-Gómez et al., 2022 (10.3390/cells11020227), who provide compelling evidence that, while IDE homologs are indeed widespread across Archaea and Eukarya, their subcellular localization and possibly their roles have evolved significantly. Notably, the authors show that bacterial IDE homologs typically contain signal peptides directing them to the secretory pathway, whereas eukaryotic IDE lacks such sequences and functions predominantly in the cytosol. This evolutionary shift in localization may reflect functional diversification beyond insulin degradation. Additionally, their findings in microglia demonstrate dynamic localization of IDE to multivesicular bodies and its release in extracellular vesicles upon activation, suggesting a broader context for IDE function and subcellular localization that merits discussion.
IDE and insulin target
- The role of IDE in insulin degradation under physiological conditions remains debated. While some studies have observed that mice lacking IDE exhibit elevated plasma insulin levels, others have reported no difference compared to wildtype controls (as reviewed by Gonzalez-Casimiro et al., 2021. Please discuss with more depth the current evidence.
- Since IDE is mostly cytosolic the mechanisms by which IDE accesses each substrate are still poorly understood, as reviewed by Leissring, 2021 (10.3390/cells10092445). IDE association with plasma membrane has already in hepatocytes (Duckworth, 1979), skeletal muscle (Yokono et al., 1979) neurons (Bulloj et al., 2008; Vekrellis et al., 2000) and microglia (Corraliza-Gomez et al., 2022). In addition, it has been reported that IDE partitions between soluble and membrane fractions in both cell lines and primary glial cultures, which is in accordance with the existence of at least two pools of cellular IDE: the cytosolic one, with a longer half-life, and the membrane-associated, with a faster turnover (Bulloj et al., 2008). Nevertheless, contrary to previous works that detected IDE on the cell surface (Bulloj et al., 2008; Goldfine et al., 1984; Yokono et al., 1982), in Corraliza-Gomez et al., 2022 reported that IDE is associated with membranes at the cytosolic side, but never on the cell surface, which makes sense with the bioinformatic analyses presented in the same study, since the fact of not having a signal peptide is an argument against IDE being able to reach the cell surface. Furthermore, a small proportion of IDE has been reported to be present in particular types of membrane microdomains, mainly lipid rafts (Bulloj et al., 2008; Corraliza-Gomez et al., 2022). This information should be discussed by the authors.
- I have some concerns regarding insulin degradation by IDE in endosomes: IDE is known to be most active at neutral to slightly basic pH, with an optimal range of pH 7.3–8.5. However, endosomes, particularly late endosomes and lysosomes, are more acidic, with pH values typically around 5.0–6.0. This pH range is suboptimal for IDE activity, and several studies have confirmed that IDE's catalytic efficiency drops significantly at lower pH (Duckworth et al., 1998; Vekrellis et al., 2000). Please discuss this.
IDE and non-insulin target
- Is the reference 27 (Duckworth, 1998) correct for the sentence “IDE performs several regulatory functions, including the modulation of androgen and glucocorticoid receptors, participating in peroxisomal fatty acid oxidation, facilitating antigen presentation, and supporting cellular growth and differentiation, as well as being involved in proteasomal degradation”? As far as I know many of these non-proteolytic functions have been described recently and the reference used is almost 30 years old. Please check.
- “For example, administration of 6bK to subjects with T2DM led to increased levels of these two molecules during the intraperitoneal glucose tolerance test, suggesting that IDE inhibitors may alter the insulin-glucagon ratio” This sentence is very inaccurate. “Subjects” seems human patients, and in the study by Sanz-Gonzalez et al., 2023, they used a HFD-induced preclinical model of T2D to test the effects of an IDE activator (PIF) and an IDE pharmacological inhibitor (6bK). “increased levels of these two molecules” which molecules? In fact, the main conclusion of the cited study is not about 6bK but the IDE activator PIF instead, suggesting that PIF activates IDE in pancreatic ß-cells and thereby increases insulin secretion.
IDE and ß-amyloid target
- Aß species accumulate in the brain in a wide variety of conformations, not only plaques but also monomers, oligomers and fibrils, and all these forms have a significant role in neuroinflammation.
- As stated before, the are several references in the literature, both reviews and primary research, hypothesizing and discussing where can IDE meet and degrade Aß. Since this review, according to its title, is focused on IDE’s role in Aß management, this section deserves further discussion and more references than just “IDE is one of the enzymes responsible for breaking down Aβ[32]. It is found in the brain, and is believed to help clear Aβ from the extracellular space[33]”.
- In my opinion, to address the role of IDE in brain Aß degradation, authors should discuss the literature available on IDE transgenic models (knock-in and knockout mice, primary cells from these animals, shRNA knockdown…), as well as Goto-Kakizaki rats, an spontaneous model of T2D which harbors mutations in the IDE gene. Several relevant references about this aspect are missing here: Farris et al., 2003; Leissring et al., 2003; Farris et al., 2004; Corraliza-Gomez et al., 2023; Morito et al., 2025; Feng et al., 2025… Since this part is the core topic of the review, it deserves more depth.
- Regarding the connections between AD-T2DM, authors might discuss some population-based human studies linking AD, T2D and IDE, for example: Kulstad et al., 2005 (demonstrated that AD patients respond differently to insulin treatment in comparison with healthy controls: while levels of plasma Aβ decreased in healthy controls after receiving a peripheral insulin infusion, the AD patients showed an increase in Aβ levels in a dose-dependent manner); Caccamo et al., 2005 (the levels of IDE have been found to decline as a result of aging in certain brain areas susceptible to AD pathology, such as the hippocampal regions); Perez et al., 2000 (the activity of IDE was significantly lower in AD brains compared with controls); Cook et al., 2003 (decreased levels of IDE mRNA and protein in hippocampal regions of AD brains compared with healthy brains).
- The manuscript relies heavily on review articles to support key biological claims. While reviews are valuable for providing overviews, primary research articles should be prioritized when referencing specific mechanistic data or experimental findings. This ensures that the cited evidence is accurate, traceable, and based on original observations.
- Figure 1: how would authors relate systemic hyperinsulinemia with brain insulin-resistance? How is the mechanism for insulin transport across the brain? And what about the evidence on target cell-specific insulin receptor knockout mice (NIRKO, MG-IRKO, etc)? The figure needs to be supported by evidence on the literature.
Genetic and structural considerations
- Provided the title of this section, I would expect a discussion about genetic studies supporting IDE as a candidate gene for both T2D and AD. In fact, several papers have reported evidence that the region in and/or around IDE might be genetically associated with both AD (Kehoe et al., 1999; Myers, 2000; Bertram, 2000; Bian et al., 2004; Liu et al., 2007; Hamshere et al., 2007; Muller et al., 2007; Vepsalainen et al., 2007; Björk et al., 2007; Zuo and Jia, 2009; Carrasquillo et al., 2010) and T2DM (Duggirala et al., 1999; Wiltshire et al., 2001; Karamohamed et al., 2003; Furukawa et al., 2008). Furthermore, IDE has been linked to other quantitative indices of AD, such as plasma Aβ levels, memory and cognitive function, CSF levels of tau protein, cerebral Aβ and neurofibrillary tangles load (Liu et al., 2007; Muller et al., 2007; Prince et al., 2003).All these references, and probably many more, are missing here.
Preclinical findings
- As stated above, the discussion should be based on primary research articles more than in reviews. As stated by Gonzalez-Casimiro et al., 2021, besides total IDE-KO mice, where IDE function is totally absent, there are some organ-specific IDE KO mice, i.e. liver-IDE-KO, alpha cells-IDE-KO, ß-cells-IDE-KO… authors should discuss briefly the phenotypes observed in each model, also considering the exportation of IDE in extracellular vesicles.
- Regarding the roles of IDE in glial cells, the references Corraliza-Gomez et al., 2023 and Yamamoto et al., 2018 deserve further discussion here.
- Particularly in microglia phenotypes, (Heneka et al., 2013; Heneka et al., 2014) already suggested that IDE might be involved in switching microglia into an anti-inflammatory phenotype, and also Corraliza-Gomez et al., proposed that IDE modulates microglial phenotype through non-enzymatic functions.
- Please discuss the relative relevance and/or primary role of IDE and neprilysin in degrading Aß.
- This section is a bit confusing to me since it jumps from models to treatments to diets… I would suggest to rearrange it to clearly separate different classes of evidence.
Clinical evidence
- The section starts with “numerous clinical and epidemiological studies” but then only a paper about this is cited. Please give more depth to this part.
- It is not the formation of Aß, but the accumulation of Aß species the main pathological hallmark for AD. If Aß was formed through cleavage of APP but then it was removed by glial cells, through the glymphatic system or by any other mechanism, there would be no Aß accumulation.
- This sentence seems out of context: “Other studies also support that T2DM promotes neurodegeneration, particularly tau pathology, but its direct effect on amyloid plaque load is less consistent [72]”. What does this add to IDE’s role?
Therapeutic perspectives
- Sentence in lines 317-319 is the same as the one in lines 310-312.
- The paragraph suggests that innovative therapeutic strategies are focusing on activating IDE or inhibiting its insulin-binding capacity to promote Aβ clearance, but several aspects require clarification and caution. First, the pharmacological activation of IDE remains largely theoretical, with very few compounds showing robust efficacy or selectivity in vivo. Are there concrete examples of small-molecule IDE activators currently under development or in preclinical testing? The mention of a synthetic PIF-derived peptide is intriguing, but the mechanism by which it modulates IDE specificity toward Aβ over insulin is not fully elucidated, and the implications of competitively displacing insulin from IDE are potentially problematic.
- Moreover, the idea of inhibiting insulin binding to IDE to favor Aβ degradation appears counterintuitive from a metabolic standpoint. Insulin is a major IDE substrate, and interfering with its clearance may increase circulating insulin levels, potentially aggravating insulin resistance and related pathologies, especially in elderly or metabolically compromised patients. Although the authors appropriately note this concern in the final sentence, it may be beneficial to further emphasize the delicate balance between enhancing Aβ clearance and preserving metabolic homeostasis, and to specify what kinds of strategies might mitigate such risks.
- Finally, the paragraph would be strengthened by distinguishing between proof-of-concept cell culture findings and clinically translatable therapeutic approaches, and by citing more recent or comprehensive studies on IDE-targeting compounds, including their limitations and safety profiles.
Conclusions
- “In conclusion, multiple factors can alter the expression or activity of IDE, including neuroinflammation, high-fat diets, genetic polymorphisms, oxidative stress, and brain aging.” I do not feel this has been addressed in the review.
- Since this review tries to link AD and T2D through IDE, it would be relevant to have any mention to human islet APP (hiAPP), which accumulates in the pancreas in diabetic conditions and, in theory, it can be also degraded by IDE.
- Figure 2 could be enhanced by adding more details, for example references supporting each factor, symbols indicating whether each factor increases or decreases IDE expression/activity or even a short sentence as a take-home message.
- I disagree with the conclusion by several reasons:
- The concluding statement suggests that clinical findings increasingly support IDE as a molecular bridge between insulin metabolism and amyloid pathology, and that selective modulation of IDE could offer a dual therapeutic benefit. While this is an appealing hypothesis, it may currently overstate the translational readiness of IDE-targeted strategies. Clinical evidence directly linking IDE modulation to improvements in both metabolic and cognitive outcomes remains limited and somewhat indirect. Most data supporting this dual role derive from preclinical models or in vitro studies, which do not always translate to clinical efficacy or safety.
- The idea of “selective modulation” of IDE implies a level of substrate specificity that is not yet achievable with current pharmacological tools. IDE is a highly promiscuous enzyme, and efforts to modulate it risk unintended effects on a wide array of physiological peptides, including glucagon, amylin, and others. The paragraph would benefit from acknowledging these challenges, and from clarifying what “selective modulation” might entail in practical terms. In addition, modulators of IDE activity had a tiny place in the review, and would deserve more discussion.
- The call for further research is appropriate, but the suggestion that IDE-targeted therapies could serve as personalized treatments for both metabolic and neurodegenerative diseases may be premature without more robust human data. A more cautious and nuanced conclusion would better reflect the current state of the field.
Author Response
Reviewer 2
Cerasuolo et al. have reviewed IDE’s role in regulating insulin and degrading Aβ, highlighting its relevance in T2DM and AD and proposing its dual role as a therapeutic target. While the review offers a general overview, several key issues limit its quality. The claim that IDE degrades Aβ plaques is misleading, as IDE’s activity has only been shown against soluble Aβ monomers in vitro, with its in vivo role still debated. The manuscript lacks critical evaluation of controversial hypotheses, such as the “competition hypothesis” between insulin and Aβ for IDE, which lacks physiological relevance given brain insulin levels and substrate affinities. It also relies heavily on secondary sources, neglecting important primary studies, including genetic and in vivo research crucial to understanding IDE in AD and diabetes. Key concepts like IDE’s subcellular localization, presence in extracellular vesicles, and pH-dependent activity are missing or superficially addressed, reducing the manuscript’s depth and translational value. Overall, important mechanistic and genetic studies are insufficiently discussed or omitted, leading to an incomplete and oversimplified perspective.
I thank the authors for their efforts but note that significant issues must be addressed before publication. If these are comprehensively corrected—by revising overstatements, citing primary literature, including missing studies, and offering a balanced critical analysis—I recommend major revision. Otherwise, the manuscript is unsuitable for publication in its current form.
Thank you for your feedback. We greatly appreciate your thorough review and constructive comments. We will address all the issues you raised with the utmost care. Specifically, we will revise any overstatements, ensure proper citation of primary literature, include the relevant missing studies, and provide a more balanced and critical analysis throughout the manuscript. A revised version will be submitted that comprehensively reflects these improvements.
COMMENTS
1.Abstract
The affirmation that IDE can break amyloid beta plaques is not correct.
The seminal paper about IDE’s ability to degrade amyloid-ß peptides (Kurochkin and Goto, 1994; 1016/0014-5793(94)00387-4) was performed in vitro and assessed the degradation of synthetic monomeric Aß peptides Aß1-40 and Aß1-28.
In Farris et al, 2003 (10.1073/pnas.0230450100), authors demonstrated Aß1-40 degradation deficits in soluble and brain membrane fractions. The work by Miller et al 2003 (10.1073/pnas.1031520100) reported increased Aß40 and Aß42 peptides accumulation in the brains from IDE deficient mice in an IDE dose-dependent manner.
A very recent paper by Morito et al, 2025 (10.1523/JNEUROSCI.2152-24.2025) reports that neprilysin deficiency accelerates Aß plaque formation more prominently than IDE deficiency.
The role of IDE in degrading Aß in vivo is questionable, as discussed by Corraliza-Gomez et al, 2023 (10.1186/s12974-023-02914-7). As discussed in that paper, a major question that needs to be addressed is where can IDE interact with Aß, since the protease and its putative substrate are localized in different subcellular compartments.
The “competition hypothesis” between insulin and Aß for IDE’s protease activity is questionable. IDE degrades insulin with higher affinity than Aß (Km ~100 nM vs >2 μM, respectively (Hoyer, 2006), but insulin levels in the brain are ~3.8 pM (Geijselaers et al., 2017; 10.3233/JAD-170522), much lower than the insulin Km, and thus unlikely to competitively inhibit Aβ degradation. Therefore, it was suggested already in 2016 that the hypothesis about competitive inhibition of IDE needs to be reformulated (Pivovarova et al., 2016; 10.1080/07853890.2016.1197416).
1.Thank you for your valuable feedback. We have thoroughly revised the abstract to better highlight these key concepts, as shown below (See lines 25-42).
“Insulin-Degrading Enzyme (IDE) plays a critical role in regulating insulin levels in various tissues, including the brain, liver, and kidneys. In Type 2 Diabetes Mellitus (T2DM), key features include insulin resistance, elevated insulin levels in the blood, and hyperglycemia. In this context, the function of IDE becomes particularly important; however, in T2DM, IDE's function can be impaired. Notably, individuals with T2DM have a higher risk of developing Alzheimer's disease (AD), suggesting that impaired IDE function may contribute to both diabetes and neurodegeneration. IDE has been studied for its ability to degrade amyloid-β (Aβ) peptides, the primary constituents of amyloid plaques in AD. However, its role in Aβ clearance in vivo remains debated due to limited enzymatic efficacy under physiological conditions and differences in subcellular localization between IDE and its putative substrate. Other proteases, such as neprilysin, appear to play a more prominent role in preventing plaque formation. Additionally, the long-standing hypothesis that insulin competes with Aβ for IDE activity has been questioned, as brain insulin levels are too low to inhibit Aβ degradation significantly. Genetic variants in the IDE gene have been associated with increased AD risk, although the mechanisms by which they alter enzyme function are not yet fully understood. A deeper understanding of IDE’s role in the context of both metabolic and neurodegenerative diseases may provide valuable insights for the development of new therapeutic strategies.”
2.Introduction
It is worth noting that the reported substrates degraded by IDE have been primarily identified in vitro using biochemical or cell-free systems, which may not fully reflect physiological relevance (Duckworth et al., 1998; doi:10.1210/edrv.19.5.0349; Shen et al., 2006; doi:10.1038/nature05143)
Regarding the last sentence of the introduction, although IDE has been implicated in Aβ catabolism, the notion that increasing IDE levels in the CNS would straightforwardly improve Aβ clearance and cognitive outcomes is oversimplified and not fully supported by in vivo evidence. In fact, IDE degrades a wide variety of substrates, including insulin, amylin, and other bioactive peptides, often with higher affinity than for Aβ (Farris et al., 2003; doi.org/10.1073/pnas.0230450100; Gonzalez-Casimiro et al., 2021; 10.3390/biomedicines9010086). This substrate promiscuity raises concerns that increasing IDE activity might disrupt other regulatory pathways, particularly those involved in insulin signaling.
2.Thank you for the suggestions on this matter. We have revised the introduction accordingly.
“However, it is important to note that many of these reported substrates have been primarily identified through in vitro assays or cell-free systems, which may not accurately reflect physiological relevance[12,13]. While IDE has been implicated in Aβ catabolism, the idea that increasing IDE levels in the central nervous system (CNS) would directly enhance Aβ clearance and improve cognitive function remains speculative. In vivo evidence does not fully support this hypothesis. Moreover, IDE exhibits high substrate promiscuity and often shows greater affinity for peptides such as insulin and amylin than for Aβ[14,15]. This raises concerns that augmenting IDE activity could interfere with other essential regulatory pathways, particularly those related to insulin signaling. IDE may contribute to the regulation of Aβ accumulation in Alzheimer’s disease (AD) and diabetic cognitive impairment. While some studies have proposed that increasing IDE levels in the central nervous system (CNS) might facilitate Aβ degradation and offer cognitive benefits, this hypothesis remains under investigation. Further exploration is required to determine the extent to which IDE influences these processes and whether modulating its activity could have therapeutic implications[11].”
3.IDE activity
While the structural description of IDE is accurate, the authors may consider briefly expanding on its enzymatic activity and regulatory potential to provide a more complete overview. IDE is a zinc-dependent metalloprotease with broad substrate specificity, playing key roles in the degradation of insulin, amyloid-β, and other bioactive peptides. Its activity is modulated by substrate conformation, subcellular localization, and interactions with small-molecule effectors. Recent studies have identified several pharmacological modulators of IDE, including both inhibitors and activators, although their substrate selectivity and therapeutic applicability remain under investigation. Including a concise mention of these aspects would better contextualize the significance of IDE’s structure-function relationship.
3.Thank you. We have expanded the paragraph accordingly. Some of the proposed concepts have also been further addressed throughout the manuscript (See lines 478-506).
4.The authors note that IDE is evolutionarily conserved from bacteria to eukaryotes, implying functional preservation. I recommend expanding this point with reference to Corraliza-Gómez et al., 2022 (10.3390/cells11020227), who provide compelling evidence that, while IDE homologs are indeed widespread across Archaea and Eukarya, their subcellular localization and possibly their roles have evolved significantly. Notably, the authors show that bacterial IDE homologs typically contain signal peptides directing them to the secretory pathway, whereas eukaryotic IDE lacks such sequences and functions predominantly in the cytosol. This evolutionary shift in localization may reflect functional diversification beyond insulin degradation. Additionally, their findings in microglia demonstrate dynamic localization of IDE to multivesicular bodies and its release in extracellular vesicles upon activation, suggesting a broader context for IDE function and subcellular localization that merits discussion.
4.Thank you. We have expanded the paragraph accordingly. Some of the proposed concepts have also been further addressed throughout the manuscript (See lines 299-304).
However, recent evidence highlights that, despite its widespread conservation, IDE has undergone significant evolutionary adaptations. Specifically, bacterial IDE homologs often contain signal peptides that direct them toward the secretory pathway, whereas eukaryotic IDE typically lacks such sequences and localizes predominantly in the cytosol. This shift in subcellular localization is thought to reflect a functional diversification beyond insulin degradation. In microglia, for instance, IDE dynamically localizes to multivesicular bodies and is released in extracellular vesicles upon activation, suggesting broader roles in intercellular signaling and proteostasis regulation[19].
IDE is ubiquitously expressed, both in insulin-sensitive and non-insulin-sensitive cells, supporting a multifunctional role for this protein[20].
It is mainly localized in the cytoplasm of various organs, including the brain, heart, liver, pancreas, and muscle[21]. Additionally, IDE can be secreted via exosomes, entering the extracellular space to interact with insulin and other substrates[22]. IDE is a zinc-dependent metalloprotease characterized by broad substrate specificity, and it plays key roles in the degradation of bioactive peptides such as insulin, amyloid-β, islet amyloid polypeptide, and glucagon. Its enzymatic activity is regulated by factors such as substrate conformation, subcellular compartmentalization, and interaction with small-molecule effectors. Recent studies have also identified several pharmacological modulators of IDE, including both inhibitors and activators; however, their substrate selectivity and potential clinical applications remain under active investigation[23–25]. Including these aspects highlights the complex structure-function relationship of IDE and its relevance beyond insulin metabolism.
5.IDE and insulin target
The role of IDE in insulin degradation under physiological conditions remains debated. While some studies have observed that mice lacking IDE exhibit elevated plasma insulin levels, others have reported no difference compared to wildtype controls (as reviewed by Gonzalez-Casimiro et al., 2021. Please discuss with more depth the current evidence.
5.Thank you for the suggestion. We have expanded the concept by referring to the proposed reference (See lines 116-124).
6.Since IDE is mostly cytosolic the mechanisms by which IDE accesses each substrate are still poorly understood, as reviewed by Leissring, 2021 (10.3390/cells10092445). IDE association with plasma membrane has already in hepatocytes (Duckworth, 1979), skeletal muscle (Yokono et al., 1979) neurons (Bulloj et al., 2008; Vekrellis et al., 2000) and microglia (Corraliza-Gomez et al., 2022). In addition, it has been reported that IDE partitions between soluble and membrane fractions in both cell lines and primary glial cultures, which is in accordance with the existence of at least two pools of cellular IDE: the cytosolic one, with a longer half-life, and the membrane-associated, with a faster turnover (Bulloj et al., 2008). Nevertheless, contrary to previous works that detected IDE on the cell surface (Bulloj et al., 2008; Goldfine et al., 1984; Yokono et al., 1982), in Corraliza-Gomez et al., 2022 reported that IDE is associated with membranes at the cytosolic side, but never on the cell surface, which makes sense with the bioinformatic analyses presented in the same study, since the fact of not having a signal peptide is an argument against IDE being able to reach the cell surface. Furthermore, a small proportion of IDE has been reported to be present in particular types of membrane microdomains, mainly lipid rafts (Bulloj et al., 2008; Corraliza-Gomez et al., 2022). This information should be discussed by the authors.
6.We have discussed these aspects in the chapter. Thank you (See lines 124-144).
7.I have some concerns regarding insulin degradation by IDE in endosomes: IDE is known to be most active at neutral to slightly basic pH, with an optimal range of pH 7.3–8.5. However, endosomes, particularly late endosomes and lysosomes, are more acidic, with pH values typically around 5.0–6.0. This pH range is suboptimal for IDE activity, and several studies have confirmed that IDE's catalytic efficiency drops significantly at lower pH (Duckworth et al., 1998; Vekrellis et al., 2000). Please discuss this.
7.We have incorporated this concept. Thank you (See lines 145-155).
8.IDE and non-insulin target
Is the reference 27 (Duckworth, 1998) correct for the sentence “IDE performs several regulatory functions, including the modulation of androgen and glucocorticoid receptors, participating in peroxisomal fatty acid oxidation, facilitating antigen presentation, and supporting cellular growth and differentiation, as well as being involved in proteasomal degradation”? As far as I know many of these non-proteolytic functions have been described recently and the reference used is almost 30 years old. Please check.
8.Thank you for pointing that out — it was an insertion error. We have corrected it and used a more recent reference(See line 165).
9.“For example, administration of 6bK to subjects with T2DM led to increased levels of these two molecules during the intraperitoneal glucose tolerance test, suggesting that IDE inhibitors may alter the insulin-glucagon ratio” This sentence is very inaccurate. “Subjects” seems human patients, and in the study by Sanz-Gonzalez et al., 2023, they used a HFD-induced preclinical model of T2D to test the effects of an IDE activator (PIF) and an IDE pharmacological inhibitor (6bK). “increased levels of these two molecules” which molecules? In fact, the main conclusion of the cited study is not about 6bK but the IDE activator PIF instead, suggesting that PIF activates IDE in pancreatic ß-cells and thereby increases insulin secretion.
9.Thank you for pointing that out. We have corrected and revised the concept in the chapter (See lines 169-173).
IDE and ß-amyloid target
10.Aß species accumulate in the brain in a wide variety of conformations, not only plaques but also monomers, oligomers and fibrils, and all these forms have a significant role in neuroinflammation.
10.We have incorporated the concept. Thank you (See lines 175-178)
11.As stated before, the are several references in the literature, both reviews and primary research, hypothesizing and discussing where can IDE meet and degrade Aß. Since this review, according to its title, is focused on IDE’s role in Aß management, this section deserves further discussion and more references than just “IDE is one of the enzymes responsible for breaking down Aβ[32]. It is found in the brain, and is believed to help clear Aβ from the extracellular space[33]”.
11.Thank you for the valuable input. The various points raised have been addressed throughout the manuscript, with expanded and detailed discussion of the underlying mechanisms. We have also incorporated numerous additional references, including both preclinical and clinical studies, to support and contextualize the revised sections more comprehensively.
12.In my opinion, to address the role of IDE in brain Aß degradation, authors should discuss the literature available on IDE transgenic models (knock-in and knockout mice, primary cells from these animals, shRNA knockdown…), as well as Goto-Kakizaki rats, an spontaneous model of T2D which harbors mutations in the IDE gene. Several relevant references about this aspect are missing here: Farris et al., 2003; Leissring et al., 2003; Farris et al., 2004; Corraliza-Gomez et al., 2023; Morito et al., 2025; Feng et al., 2025… Since this part is the core topic of the review, it deserves more depth.
12.Thank you for this important point of reflection. The preclinical studies have been thoroughly discussed in the manuscript chapter titled: “4. PRECLINICAL FINDINGS ON THE RELATIONSHIP BETWEEN IDE, INSULIN, AND Aβ.”
13.Regarding the connections between AD-T2DM, authors might discuss some population-based human studies linking AD, T2D and IDE, for example: Kulstad et al., 2005 (demonstrated that AD patients respond differently to insulin treatment in comparison with healthy controls: while levels of plasma Aβ decreased in healthy controls after receiving a peripheral insulin infusion, the AD patients showed an increase in Aβ levels in a dose-dependent manner); Caccamo et al., 2005 (the levels of IDE have been found to decline as a result of aging in certain brain areas susceptible to AD pathology, such as the hippocampal regions); Perez et al., 2000 (the activity of IDE was significantly lower in AD brains compared with controls); Cook et al., 2003 (decreased levels of IDE mRNA and protein in hippocampal regions of AD brains compared with healthy brains).
13.Thank you for this valuable point of reflection. The clinical studies have been thoroughly discussed in the manuscript chapter titled: “5. CLINICAL EVIDENCE ON THE RELATIONSHIP BETWEEN IDE, INSULIN, AND Aβ.” We have included a detailed description of the most recent clinical studies from the literature related to this topic.
14.The manuscript relies heavily on review articles to support key biological claims. While reviews are valuable for providing overviews, primary research articles should be prioritized when referencing specific mechanistic data or experimental findings. This ensures that the cited evidence is accurate, traceable, and based on original observations.
14.Thank you for the suggestion. To improve the manuscript, we have significantly expanded the bibliographic references throughout, incorporating experimental studies, both preclinical and clinical.
15.Figure 1: how would authors relate systemic hyperinsulinemia with brain insulin-resistance? How is the mechanism for insulin transport across the brain? And what about the evidence on target cell-specific insulin receptor knockout mice (NIRKO, MG-IRKO, etc)? The figure needs to be supported by evidence on the literature.
15.Thank you for the critical insight. We have revised the caption of Figure 1 to better clarify the concept and added the relevant references. We have also revised the image, as suggested by the other reviewer.
Figure 1. Insulin-Degrading Enzyme (IDE) Activity and Aβ Degradation.
(A) Under physiological conditions, IDE maintains brain homeostasis by degrading amyloid-beta (Aβ) peptides, preventing their accumulation. (B) In states of systemic hyperinsulinemia, such as type 2 diabetes, IDE activity is reduced due to substrate competition and metabolic dysregulation, leading to impaired Aβ clearance and plaque accumulation. In parallel, chronic hyperinsulinemia impairs insulin transport across the blood–brain barrier via receptor downregulation and induces central insulin resistance through intracellular signaling desensitization. These combined mechanisms exacerbate Aβ buildup and neurodegeneration.
Farris, W.; Mansourian, S.; Chang, Y.; Lindsley, L.; Eckman, E.A.; Frosch, M.P.; Eckman, C.B.; Tanzi, R.E.; Selkoe, D.J.; Guenette, S. Insulin-Degrading Enzyme Regulates the Levels of Insulin, Amyloid Beta-Protein, and the Beta-Amyloid Precursor Protein Intracellular Domain in Vivo. Proc Natl Acad Sci U S A 2003, 100, 4162–4167, doi:10.1073/pnas.0230450100.
González-Casimiro, C.M.; Merino, B.; Casanueva-Álvarez, E.; Postigo-Casado, T.; Cámara-Torres, P.; Fernández-Díaz, C.M.; Leissring, M.A.; Cózar-Castellano, I.; Perdomo, G. Modulation of Insulin Sensitivity by Insulin-Degrading Enzyme. Biomedicines 2021, 9, 86, doi:10.3390/biomedicines9010086.
Genetic and structural considerations
16.Provided the title of this section, I would expect a discussion about genetic studies supporting IDE as a candidate gene for both T2D and AD. In fact, several papers have reported evidence that the region in and/or around IDE might be genetically associated with both AD (Kehoe et al., 1999; Myers, 2000; Bertram, 2000; Bian et al., 2004; Liu et al., 2007; Hamshere et al., 2007; Muller et al., 2007; Vepsalainen et al., 2007; Björk et al., 2007; Zuo and Jia, 2009; Carrasquillo et al., 2010) and T2DM (Duggirala et al., 1999; Wiltshire et al., 2001; Karamohamed et al., 2003; Furukawa et al., 2008). Furthermore, IDE has been linked to other quantitative indices of AD, such as plasma Aβ levels, memory and cognitive function, CSF levels of tau protein, cerebral Aβ and neurofibrillary tangles load (Liu et al., 2007; Muller et al., 2007; Prince et al., 2003).All these references, and probably many more, are missing here.
16.Thank you for the valuable advice. We have integrated the genetics section by including genetic studies that highlight the role of IDE in AD (See lines 231-250).
Among the genetic variants investigated in relation to AD, the rs1887922 polymorphism within the IDE gene has garnered attention for its potential role in modulating disease risk. A recent meta-analysis, which included four independent studies, identified a significant association between rs1887922 and increased susceptibility to AD, with odds ratios greater than 1 across four genetic models[55]. Moreover a case-control study involving 406 Han Chinese individuals from the Xinjiang region investigated the association between two IDE gene SNPs (rs1887922 and rs1999764) and LOAD risk. The CT+CC genotype of rs1999764 was found to exert a protective effect compared to the TT genotype (adjusted P = .0001; OR = 0.226), whereas the CT+CC genotype of rs1887922 was associated with a higher risk of LOAD (adjusted P = .0001; OR = 3.640). These associations were independent of the apolipoprotein E ε4 polymorphism and displayed sex-specific patterns, with rs1887922 being more relevant in men and rs1999764 in women[54].
Another study investigated the association between the rs2421943 SNP in the IDE gene and the risk of mild cognitive impairment (MCI) and AD. The study included two independent cohorts totaling 1,670 individuals (595 AD patients, 400 controls; 135 MCI patients, 540 controls). Genotypic analysis by PCR and restriction enzyme digestion revealed that AG and GG genotypes were significantly associated with an increased risk of AD, while AG was also linked to a higher risk of MCI. In silico analysis showed that rs2421943 lies within a predicted binding site for hsa-miR-7110-5p, suggesting a possible post-transcriptional regulatory mechanism that may reduce IDE levels, impair amyloid-β clearance, and thereby contribute to disease progression. These findings support rs2421943 as a genetic risk factor for both AD and its prodromal stage[56].
Regarding the second request, during the work we included studies that emphasize the link between IDE and the various aspects you requested (See lines 437-454).
Preclinical findings
17.As stated above, the discussion should be based on primary research articles more than in reviews. As stated by Gonzalez-Casimiro et al., 2021, besides total IDE-KO mice, where IDE function is totally absent, there are some organ-specific IDE KO mice, i.e. liver-IDE-KO, alpha cells-IDE-KO, ß-cells-IDE-KO… authors should discuss briefly the phenotypes observed in each model, also considering the exportation of IDE in extracellular vesicles.
17.Thank you for the suggestions. We have integrated these aspects into the manuscript along with the relevant preclinical studies (See lines 272-297).
18.Regarding the roles of IDE in glial cells, the references Corraliza-Gomez et al., 2023 and Yamamoto et al., 2018 deserve further discussion here.
18.Thank you for the insightful suggestion. We have integrated a more in-depth discussion of the role of IDE in glial cells (See lines 355-371).
.
19.Particularly in microglia phenotypes, (Heneka et al., 2013; Heneka et al., 2014) already suggested that IDE might be involved in switching microglia into an anti-inflammatory phenotype, and also Corraliza-Gomez et al., proposed that IDE modulates microglial phenotype through non-enzymatic functions.
19.These concepts have been incorporated into the manuscript. Thank you. (See lines 355-371).
20.Please discuss the relative relevance and/or primary role of IDE and neprilysin in degrading Aß.
20.We have discussed and commented on the role of IDE and neprilysin in degrading Aβ plaques (see lines 383-393).
21.This section is a bit confusing to me since it jumps from models to treatments to diets… I would suggest rearranging it to clearly separate different classes of evidence.
21.We have reorganized all the section to improve its clarity and structure; thank you for the valuable suggestion.
Clinical evidence
22.The section starts with “numerous clinical and epidemiological studies” but then only a paper about this is cited. Please give more depth to this part.
22.We have expanded this section by adding several clinical studies (See lines 414-432).
23.It is not the formation of Aß, but the accumulation of Aß species the main pathological hallmark for AD. If Aß was formed through cleavage of APP but then it was removed by glial cells, through the glymphatic system or by any other mechanism, there would be no Aß accumulation.
23.Thank you for the suggestion; we have revised the sentence to make it clearer. Below, we present the revised version (See lines 406-408).
“It saturates IDE, impairing the clearance of Aβ and promoting its accumulation and amyloid plaque formation in the brain. As IDE degrades both insulin and Aβ, it serves as a vital connection between these processes[83].”
24.This sentence seems out of context: “Other studies also support that T2DM promotes neurodegeneration, particularly tau pathology, but its direct effect on amyloid plaque load is less consistent [72]”. What does this add to IDE’s role?
24.Thank you for pointing that out; we have removed the sentence, as it could indeed confuse the reader.
Therapeutic perspectives
25.Sentence in lines 317-319 is the same as the one in lines 310-312.
25.We have reorganized the text to avoid repetitions. Thank you very much.
26.The paragraph suggests that innovative therapeutic strategies are focusing on activating IDE or inhibiting its insulin-binding capacity to promote Aβ clearance, but several aspects require clarification and caution. First, the pharmacological activation of IDE remains largely theoretical, with very few compounds showing robust efficacy or selectivity in vivo. Are there concrete examples of small-molecule IDE activators currently under development or in preclinical testing? The mention of a synthetic PIF-derived peptide is intriguing, but the mechanism by which it modulates IDE specificity toward Aβ over insulin is not fully elucidated, and the implications of competitively displacing insulin from IDE are potentially problematic.
26.Thank you for the suggestion. We have integrated the text with concrete examples of small-molecule IDE activators currently under investigation (See all the section THERAPEUTIC PERSPECTIVES: MODULAR IDE TO COUNTERACT AD)
Furthermore, we have provided a clearer explanation of the role of the synthetic PIF-derived peptide (See lines 480-485).
27.Moreover, the idea of inhibiting insulin binding to IDE to favor Aβ degradation appears counterintuitive from a metabolic standpoint. Insulin is a major IDE substrate, and interfering with its clearance may increase circulating insulin levels, potentially aggravating insulin resistance and related pathologies, especially in elderly or metabolically compromised patients. Although the authors appropriately note this concern in the final sentence, it may be beneficial to further emphasize the delicate balance between enhancing Aβ clearance and preserving metabolic homeostasis, and to specify what kinds of strategies might mitigate such risks.
27.Thank you for the important insight. We have emphasized this point further in the conclusions of the paper (See lines 531-574).
28.Finally, the paragraph would be strengthened by distinguishing between proof-of-concept cell culture findings and clinically translatable therapeutic approaches, and by citing more recent or comprehensive studies on IDE-targeting compounds, including their limitations and safety profiles.
28.We have addressed this point by explicitly clarifying the preliminary nature of the in vitro findings in the conclusions of the manuscript. Furthermore, we have expanded Section 6 by including potential therapeutic approaches aimed at modulating IDE activity.
Conclusions
29.“In conclusion, multiple factors can alter the expression or activity of IDE, including neuroinflammation, high-fat diets, genetic polymorphisms, oxidative stress, and brain aging.” I do not feel this has been addressed in the review. Figure 2 could be enhanced by adding more details, for example references supporting each factor, symbols indicating whether each factor increases or decreases IDE expression/activity or even a short sentence as a take-home message.
29.We thank the reviewer for this observation. While several factors influencing IDE expression or activity—such as neuroinflammation, high-fat diets, oxidative stress, brain aging, and genetic polymorphisms—are discussed throughout the main text, we understand the importance of making their collective role more visible. To address this, we have included a dedicated figure (now Figure 3) in the conclusion section, accompanied by a revised caption that clearly summarizes all the main factors known to reduce IDE expression and/or activity. This visual synthesis is intended to provide a concise and accessible overview of the multifactorial regulation of IDE, in line with the topics covered throughout the manuscript. Specifically, we have updated the figure by adding relevant elements and integrated appropriate supporting references within the caption: [25, 37, 85, 92, 93]. Below, we provide the updated caption.
“Figure 3. Pathophysiological and environmental factors influencing IDE activity.
Multiple elements—including neuroinflammation, amyloid-β (Aβ) plaque accumulation, insulin resistance, infectious pathogens, brain aging, genetic polymorphisms, oxidative stress, dysregulation of metal ions (Fe²⁺/Fe³⁺, Zn²⁺, Cu²⁺, Al³⁺), and high-fat diet—contribute to reduced IDE expression or enzymatic activity. This impairment hinders the degradation of insulin and Aβ, promoting neurodegenerative and metabolic processes characteristic of type 2 diabetes and Alzheimer’s disease.”
30.Since this review tries to link AD and T2D through IDE, it would be relevant to have any mention to human islet APP (hiAPP), which accumulates in the pancreas in diabetic conditions and, in theory, it can be also degraded by IDE.
30.They are mentioned at lines 67–69. Thank you.
31.I disagree with the conclusion by several reasons:
The concluding statement suggests that clinical findings increasingly support IDE as a molecular bridge between insulin metabolism and amyloid pathology, and that selective modulation of IDE could offer a dual therapeutic benefit. While this is an appealing hypothesis, it may currently overstate the translational readiness of IDE-targeted strategies. Clinical evidence directly linking IDE modulation to improvements in both metabolic and cognitive outcomes remains limited and somewhat indirect. Most data supporting this dual role derive from preclinical models or in vitro studies, which do not always translate to clinical efficacy or safety.
The idea of “selective modulation” of IDE implies a level of substrate specificity that is not yet achievable with current pharmacological tools. IDE is a highly promiscuous enzyme, and efforts to modulate it risk unintended effects on a wide array of physiological peptides, including glucagon, amylin, and others. The paragraph would benefit from acknowledging these challenges, and from clarifying what “selective modulation” might entail in practical terms. In addition, modulators of IDE activity had a tiny place in the review, and would deserve more discussion.
The call for further research is appropriate, but the suggestion that IDE-targeted therapies could serve as personalized treatments for both metabolic and neurodegenerative diseases may be premature without more robust human data. A more cautious and nuanced conclusion would better reflect the current state of the field.
31.We thank the reviewer for their thoughtful and well-founded comments regarding the conclusions of our manuscript. We have carefully considered the concerns raised, and we agree that our previous concluding statement may have conveyed a level of clinical readiness that is not yet fully supported by current evidence.
Accordingly, we have revised and rephrased the full text and the conclusion to better reflect the current limitations and uncertainties surrounding IDE-targeted strategies. In particular, we now acknowledge the indirect nature of most clinical data, the translational gap between preclinical findings and therapeutic applications, and the challenges related to IDE’s broad substrate specificity. We have also clarified what is meant by “selective modulation” in conceptual terms, emphasizing that this remains an aspirational goal rather than a current pharmacological reality. Furthermore, in response to the reviewer’s comment, we have expanded the discussion on IDE modulators to provide a more balanced overview of their current status and therapeutic potential. We appreciate the reviewer’s input, which has helped us to improve the clarity and accuracy of our conclusions.

Round 2
Reviewer 2 Report
Comments and Suggestions for Authors
Cerasuolo et al. have performed a significant effort in addressing my comments and concerns about the first version of the manuscript. Some minor details are presented below:
Introduction
- The addition of the notion about repurposing antidiabetic drugs for Alzheimer’s disease is a real relevant point for the topic. However, the mention to metformin inhibitory activity on MARK4 seems a bit disconnected, even in a separate paragraph (lines 58-62). Authors may insert this in previous paragraph as an example of the repurposing of antidiabetics in neurodegenerative diseases.
- Alzheimer’s disease abbreviation is introduced in line 80 but “Alzheimer’s disease” appears early in the text (line 57) and it should be introduced there.
IDE activity
- “The cytoplasm of various organs” in line 102 does not seem correct to me. Authors should say in the cytoplasm of different cell types in various organs or something similar, since the organs do not have cytoplasms but different cell types with cytoplasms instead.
IDE and insulin target
- Line 134: Reference 19 does not support IDE on the external cell surface, but associated to the membrane by its cytosolic side instead. Other previous studies (Bulloj et al., 2008; Yokono et al., 1982; Goldfine er al., 1984) report such external cell surface localization, please cite them properly.
- The seminal reference identifying IDE in lipid rafts is from Bulloj et al., 2008 in neurons. When referring to IDE location in lipid rafts (line 139), authors should cite both reference 19 in microglia and this one from Bulloj in neurons.
IDE and ß-amyloid target
- The competition hypothesis is briefly introduced in the abstract of the manuscript, but it should appear here also. In lines 181-193, IDE is presented as the main Aß degrading enzyme and the main take home message is that “IDE becomes overwhelmed with insulin” (lines 192-193). However, as I commented in my previous review, IDE degrades insulin with higher affinity than Aß (Km ~100 nM vs >2 μM, respectively (Hoyer, 2006), but insulin levels in the brain are ~3.8 pM (Geijselaers et al., 2017; 10.3233/JAD-170522), much lower than the insulin Km, and thus unlikely to competitively inhibit Aβ degradation. Therefore, it was suggested already in 2016 that the hypothesis about competitive inhibition of IDE needs to be reformulated (Pivovarova et al., 2016; 10.1080/07853890.2016.1197416). I consider that authors should present side by side both theories, otherwise the discussion about this topic is severely biased. This is evident also in figure 1, where the main readout is that IDE degrades Aß and insulin competes with Aß for IDE’s enzymatic activity. While this is a valid theory and figure 1 is pertinent in its current form, it should be accompanied also by numbers (physiological concentrations, Km for each substrate, etc).
Therapeutic perspectives
- Lines 469-471 are identical to lines 475-477. Please fix it.
Conclusions
- Line 541: it is Figure 3, not 2.
Congratulations to the authors for their good work!
Author Response
Reviewer 2
Introduction
- The addition of the notion about repurposing antidiabetic drugs for Alzheimer’s disease is a real relevant point for the topic. However, the mention to metformin inhibitory activity on MARK4 seems a bit disconnected, even in a separate paragraph (lines 58-62).Authors may insert this in previous paragraph as an example of the repurposing of antidiabetics in neurodegenerative diseases.
Thank you for your valuable suggestion. Although we initially included the sentence regarding metformin’s inhibitory activity on MARK4 at the request of Reviewer 1, who recommended reference 4, we agree that it felt somewhat disconnected in its placement. Therefore, we have decided to remove this sentence to improve the overall coherence of the manuscript. However, we have maintained reference 4 in the manuscript.
- Alzheimer’s disease abbreviation is introduced in line 80 but “Alzheimer’s disease” appears early in the text (line 57) and it should be introduced there.
Thank you for your helpful comment. We have made the recommended correction by introducing the abbreviation AD for Alzheimer’s disease at its first occurrence (See line 54). We have ensured that the acronym AD is used consistently in all subsequent occurrences. Likewise, for type 2 diabetes, whose acronym is first introduced at line 49, we have used the acronym T2DM in all subsequent mentions.
IDE activity
- “The cytoplasm of various organs” in line 102 does not seem correct to me. Authors should say in the cytoplasm of different cell types in various organs or something similar, since the organs do not have cytoplasms but different cell types with cytoplasms instead.
Thank you for pointing this out. We agree with your observation and have revised the sentence to read “in the cytoplasm of different cell types in various organs” to improve accuracy (See line 96).
IDE and insulin target
- Line 134: Reference 19 does not support IDE on the external cell surface, but associated to the membrane by its cytosolic side instead. Other previous studies (Bulloj et al., 2008; Yokono et al., 1982; Goldfine er al., 1984) report such external cell surface localization, please cite them properly.
Thank you. We have replaced reference 19 with the three requested references, respectively Ref 30-31-32.
- The seminal reference identifying IDE in lipid rafts is from Bulloj et al., 2008 in neurons. When referring to IDE location in lipid rafts (line 139), authors should cite both reference 19 in microglia and this one from Bulloj in neurons.
Thank you for the suggestion, we have added the requested references (Ref 30-Bulloj et al)(See Line 133).
IDE and ß-amyloid target
- The competition hypothesis is briefly introduced in the abstract of the manuscript, but it should appear here also. In lines 181-193, IDE is presented as the main Aß degrading enzyme and the main take home message is that “IDE becomes overwhelmed with insulin” (lines 192-193). However, as I commented in my previous review, IDE degrades insulin with higher affinity than Aß (Km ~100 nM vs >2 μM, respectively (Hoyer, 2006), but insulin levels in the brain are ~3.8 pM (Geijselaers et al., 2017; 10.3233/JAD-170522), much lower than the insulin Km, and thus unlikely to competitively inhibit Aβ degradation. Therefore, it was suggested already in 2016 that the hypothesis about competitive inhibition of IDE needs to be reformulated (Pivovarova et al., 2016; 10.1080/07853890.2016.1197416). I consider that authors should present side by side both theories, otherwise the discussion about this topic is severely biased. This is evident also in figure 1, where the main readout is that IDE degrades Aß and insulin competes with Aß for IDE’s enzymatic activity. While this is a valid theory and figure 1 is pertinent in its current form, it should be accompanied also by numbers (physiological concentrations, Km for each substrate, etc).
Thank you for your valuable support. We have added this information along with the relevant references (See lines 187-193), and we have also included this information in the Figure 1. We have also added the sentence in the figure legend to explain better the concepts presented: “Figure 1. Insulin-degrading enzyme (IDE) Activity and Aβ Degradation. IDE plays a crucial role in maintaining the balance between the production and degradation of amyloid-beta (Aβ) peptides in the healthy brain (A). However, in cases of hyperinsulinemia, reduced IDE activity impairs the clearance of Aβ, leading to its accumulation in the brain and an increase in Aβ plaques (B). The competition hypothesis is debated. Although IDE has a higher affinity for insulin (Km ~100 μM) than for Aβ (Km >2 μM), the brain's insulin levels (~3.8 pM) are far below this threshold. In diabetic patients, the accumulation of beta-amyloid plaques leads to increased neuroinflammation, which in turn contributes to higher tau protein levels. This increase in phosphorylated tau (p-tau) is further driven by alterations in the PI3K-Akt pathway”.
Therapeutic perspectives
- Lines 469-471 are identical to lines 475-477. Please fix it.
Thank you. We have corrected.
Conclusions
- Line 541: it is Figure 3, not 2.
Thank you. We have corrected.
